# Fast synthesis of large-area bilayer graphene film on Cu

Jincan Zhang[1,2,3,4,15], Xiaoting Liu[1,2,3,15], Mengqi Zhang[2,5,15], Rui Zhang[6,15], Huy Q. Ta[7], Jianbo Sun[2], Wendong Wang[6], Wenqing Zhu[8], Tiantian Fang[9], Kaicheng Jia[1,2], Xiucai Sun[1,2], Xintong Zhang[2], Yeshu Zhu[1,2,3], Jiaxin Shao[1,2,3], Yuchen Liu[2], Xin Gao[1,2,3], Qian Yang[1,2], Luzhao Sun[1,2,3], Qin Li[2], Fushun Liang[1,2,3], Heng Chen[1,2], Liming Zheng[1,2], Fuyi Wang[10], Wanjian Yin[11], Xiaoding Wei[8], Jianbo Yin[2], Thomas Gemming[7], Mark. H. Rummeli[7,11,12,13], Haihui Liu[5,16]✉, Hailin Peng[1,2,16]✉, Li Lin[14,16]✉ & Zhongfan Liu[1,2,16]✉

Bilayer graphene (BLG) is intriguing for its unique properties and potential applications in electronics, photonics, and mechanics. However, the chemical vapor deposition synthesis of large-area high-quality bilayer graphene on Cu is suffering from a low growth rate and limited bilayer coverage. Herein, we demonstrate the fast synthesis of meter-sized bilayer graphene film on commercial polycrystalline Cu foils by introducing trace $CO_2$ during high-temperature growth. Continuous bilayer graphene with a high ratio of AB-stacking structure can be obtained within 20 min, which exhibits enhanced mechanical strength, uniform transmittance, and low sheet resistance in large area. Moreover, 96 and 100% AB-stacking structures were achieved in bilayer graphene grown on single-crystal Cu(111) foil and ultraflat single-crystal Cu(111)/sapphire substrates, respectively. The AB-stacking bilayer graphene exhibits tunable bandgap and performs well in photodetection. This work provides important insights into the growth mechanism and the mass production of large-area high-quality BLG on Cu.

Bilayer graphene (BLG) and its stacking order provide remarkable and unique properties compared to monolayer graphene (MLG), such as the tunable bandgap of AB-stacking BLG (AB-BLG)[1], the conventional superconductivity in magic-angle twisted BLG (tBLG)[2], as well as the enhanced mechanical strength[3] and electrical conductivity[4]. Therefore, the controlled preparation and applications of BLG have attracted intense academic and industrial interests[5,6]. The current methods for the synthesis of BLG mainly include mechanical exfoliation[7], artificial stacking of two individual monolayer graphene[8], and chemical vapor deposition (CVD) growth[9,10]. However, the exfoliation method suffers from small sizes of graphene flakes and therefore low production efficiency, while stacking of two monolayer graphene would inevitably introduce interfacial impurities. In contrast, the CVD approach has exhibited excellent capability to synthesize large-area,

high-quality BLG films, and the as-received BLG can satisfy the requirements of graphene-based electronic and photonic devices in terms of scalability and quality[11,12].

Despite the recent breakthrough regarding the CVD synthesis of BLG with near-100% coverage and uniform stacking order using metal alloys such as CuNi[13,14], CuSi[15], and Pt₃Si[16], the fabrication of metal alloys usually needs careful control over the amount and uniformity of alloying element in large area, which would become more difficult in the industrial-scale production. In this regard, it is still challenging to achieve the fast synthesis of large-area continuous BLG on Cu, which is the most promising metal substrate to synthesize high-quality graphene films with low cost and high uniformity[6]. The growth of adlayer graphene on Cu requires an additional supply of carbon species, and two possible routes have been reported to supply active carbon

species for growing the second-layer graphene. Carbon species can either diffuse from the edge of the MLG domains to be consumed in the nucleation and growth of second layers[17,18] or diffuse across the Cu bulk to supply the growth of the second layer on the other side of the Cu foil[19,20]. However, the reported bilayer coverage of graphene on Cu is still less than 95%[21], and several hours are required for the growth[22,23]. This is because the second-layer graphene usually nucleates underneath the first-layer graphene, resulting in the difficulty in the continuous supply of active carbon species[16]. Especially, the supply of active carbon species gradually decreases with the coverage of graphene on Cu, which would determine the amount of carbon species produced on uncovered Cu, and the growth of BLG would be immediately terminated once the Cu is fully covered by graphene[24]. The structure of Cu pockets[22] or enclosures[25] has been used for initiating the diffusion through the Cu bulk, which, however, is not compatible with mass production, for adjacent Cu would be fused together easily at high temperature. Therefore, tremendous efforts are still in great demand for achieving the fast and scalable synthesis of BLG on commercially available Cu foil.

CO$_2$, the greenhouse gas, has been widely utilized to produce valuable carbon-based nanomaterials to mitigate the adverse effects of high CO$_2$ emissions. Recently, the capability of CO$_2$ for high-temperature CVD growth of graphene has been reported by several groups[26,27]. In detail, CO$_2$ can be utilized for the pre-treatment of commercial Cu substrates based on its selective etching ability of the disordered carbon impurities[26,27] or for growing graphene as the carbon source[28]. However, when CO$_2$ is employed as a carbon source, additional catalysts, such as Ni/Al$_2$O$_3$[21,28] and CuPd[29], or special treatment of the substrates[30] are usually required, which might hinder compatibility with the mass production processes. In all, even though isolated MLG domains[28], continuous MLG films[26,27], and isolated BLG domains[21] have been successfully grown based on CO$_2$, till now, the controlled preparation of large-area high-quality continuous BLG on Cu has not been achieved yet.

Herein, we demonstrated the role of CO$_2$ in the formation of BLG and achieved the fast synthesis of continuous BLG within 20 min on the commercial polycrystalline Cu foils, which is compatible with the production of meter-sized BLG films with bilayer coverage no less than 94%. During the high-temperature growth, CO$_2$ can etch the as-formed MLG film and provide sufficient diffusion routes for carbon species to arrive at the Cu surface, which would fuel the growth of the second-layer graphene underneath. The growth mechanism was identified by an isotropic labeling technique. Transmission electron microscopy (TEM) and Raman measurements confirmed AB-stacking dominated in the as-received BLG, which also exhibits enhanced mechanical strength and reduced sheet resistance. Especially, 96 and 100% AB-stacking structure can be obtained on single-crystal Cu(111) foils and ultraflat single-crystal Cu(111)/sapphire substrates, respectively. Moreover, AB-BLG also exhibits tunable bandgap and promising performance as employed for photodetectors. This work not only provides a deeper understanding of the growth mechanism of BLG on Cu but also paves an avenue toward the mass production of high-quality BLG films for potential applications.

## Results

### Rapid growth of BLG assisted by CO$_2$

The fast synthesis of BLG film on Cu foils was carried out by introducing CO$_2$ with trace amount into the low-pressure CVD system during the high-temperature growth (Supplementary Fig. 1). After the optimization of CVD growth parameters, such as the growth pressure and gas flow rates of H$_2$ and CH$_4$, continuous BLG was obtained after 20 min growth (Supplementary Figs. 2–8). Figure 1a–c shows the representative optical microscope (OM) images of the as-prepared graphene films that were grown on polycrystalline Cu foils and then transferred onto SiO$_2$/Si substrates after varied growth times. In detail,

after 1 min growth, the whole coverage of MLG was achieved without the formation of adlayer (Fig. 1a). Notably, after the introduction of CO$_2$, the full coverage of the first-layer graphene does not restrict the growth of second-layer graphene, indicating that the growth behavior of graphene was different from the previously reported self-limited growth mechanism[24,31]. After growth for 10 min, bilayer coverage can increase to ~50% (Fig. 1b), and 20 min is sufficient for obtaining the continuous BLG (Fig. 1c and Supplementary Fig. 8).

Generally, without using CO$_2$, the final bilayer coverage of graphene is less than 10% (Supplementary Fig. 9), consistent with previous reports[24]. In contrast, the growth rate and coverage of BLG on Cu can be highly enhanced by optimizing the supply amount of CO$_2$, i.e., the flow rate and injection time of CO$_2$ accompanied by the supply of CH$_4$ for the graphene growth stage. Specifically, the BLG coverage increases linearly both with the flow rate of CO$_2$ (Fig. 1d) and with the injection time of CO$_2$ (Fig. 1e). Moreover, once the CO$_2$ is turned off, the supply of CH$_4$ and H$_2$ would not enable the increase of the bilayer coverage (Fig. 1f), indicating that continuous supply of CO$_2$ is crucial for the formation of BLG with high coverage.

In comparison with previous reports, the improvement of both BLG growth rate and coverage is clearly revealed using our CO$_2$-assisted synthesis strategy, based on which continuous BLG with an average domain size of ~30–50 μm can be obtained within 20 min (Fig. 1g). Note that the original references are also listed in Supplementary Table 1 with more details[9,10,17,21,22,32–52]. The CO$_2$-assisted strategy also exhibits excellent scalability and compatibility with the growth of large-area graphene on Cu foils. The introduction of CO$_2$ enabled the successful synthesis of eight pieces of submeter-sized BLG film (0.3 m * 0.1 m) in one batch with an average bilayer coverage higher than 95% (Fig. 1h, i and Supplementary Figs. 10–12). Moreover, the graphene film grown on a two-meter-long Cu foil with BLG coverage of ~92% was also achieved by simply introducing CO$_2$ into a homemade roll-to-roll mass production system (Supplementary Fig. 13).

### The role of CO$_2$ in facilitating BLG growth on Cu

The growth mechanism of the BLG assisted by CO$_2$ was investigated using the isotopic labeling technique. In detail, BLG films were synthesized by alternately introducing $^{13}$CH$_4$ and $^{12}$CH$_4$ every 2 min to fuel the graphene growth (Fig. 2a), while the flow rate of H$_2$ and CH$_4$ was kept constant. The graphene films after the growth of 1 min, 10 min, and 20 min were then separately transferred to SiO$_2$/Si substrates for Raman characterization, which can visualize the spatial distribution of isotopes[37,53]. Representative Raman spectra of MLG composed of $^{12}$C ($^{12}$C-MLG), BLG composed of $^{12}$C ($^{12}$C/$^{12}$C-BLG), and BLG consisting of $^{12}$C in the first layer and $^{13}$C in the second layer ($^{12}$C/$^{13}$C-BLG) are presented in Fig. 2b. The spatial distribution of $^{12}$C and $^{13}$C can be obtained from Raman intensity mappings of the G bands centered at ~1580 cm$^{-1}$ and ~1520 cm$^{-1}$, respectively, which correspond to the $^{12}$C-labeled graphene and $^{13}$C-labeled graphene. Figure 2c, d clearly reveals the formation of a continuous $^{12}$C-MLG film, which means that 1-min exposure of $^{12}$C-labeled methane is sufficient for growing fully covered MLG on Cu. The nucleation time of the second-layer graphene can be inferred by the number of observed $^{13}$C rings in one concentric hexagon and the number of introduced $^{13}$C flux, and the growth rate can be inferred from the length of $^{13}$C or $^{12}$C shell and the introduction duration of $^{13}$C or $^{12}$C-labeled methane (Fig. 2e, f, and Supplementary Fig. 14). Clearly, after the nucleation, the growth rate of BLG domains keeps nearly constant (~2 μm/min) until the coalescence with adjacent domains (Fig. 2f and Supplementary Figs. 14 and 15). In addition, the segregation growth of BLG was also excluded by the clear border between $^{12}$C-I$_G$ and $^{13}$C-I$_G$ rings.

Stacking sequence of the BLG was identified by performing mild Bi$^{3+}$ sputtering to acquire the depth profile of $^{12}$C and $^{13}$C using time-of-flight secondary ion mass spectrometry (ToF-SIMS). Note that the BLG

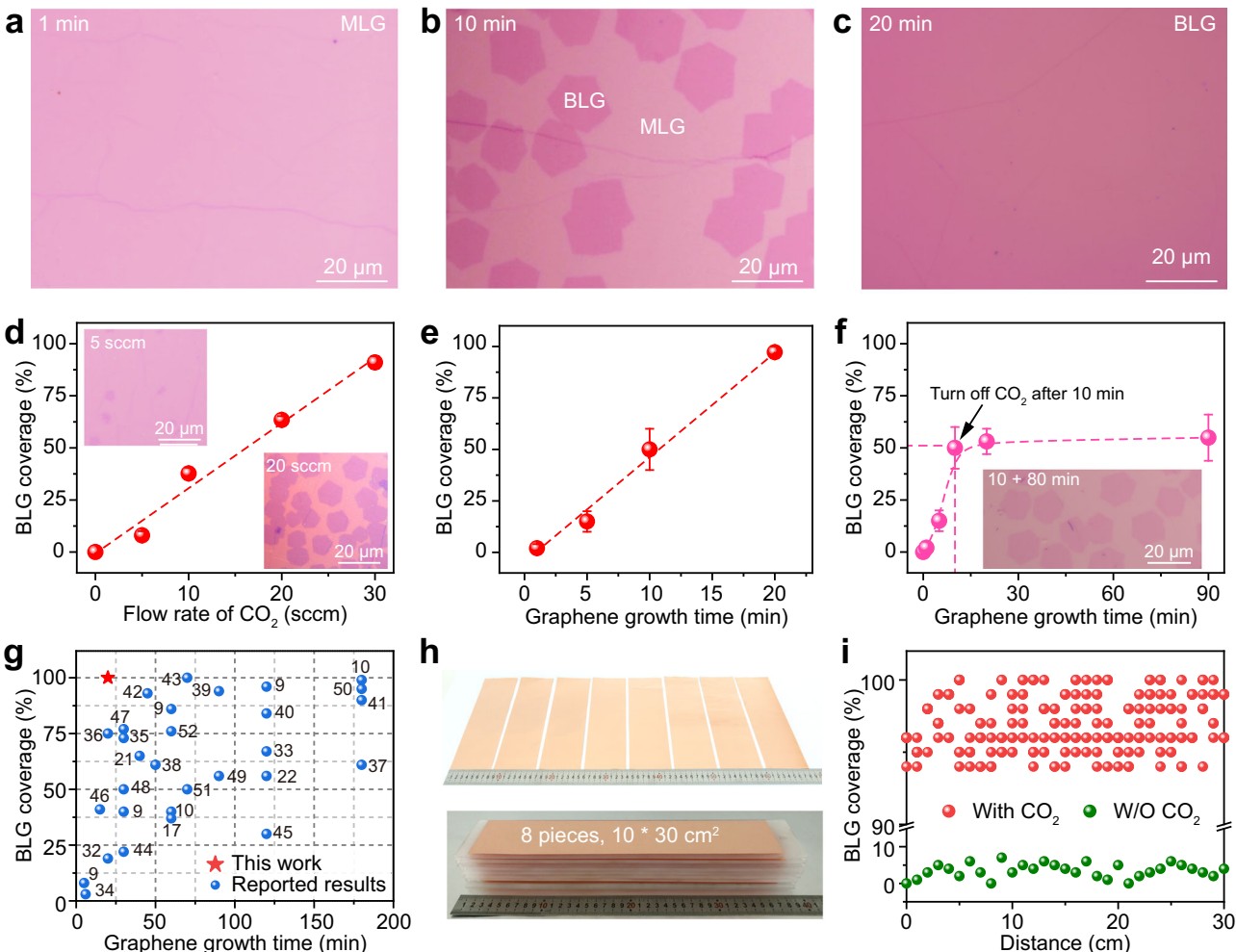

**Fig. 1 | Fast synthesis of large-area bilayer graphene (BLG) film with the assistance of $CO_2$. a−c** Optical microscope (OM) images of the transferred graphene after 1 min (**a**), 10 min (**b**), and 20 min (**c**) growth on commercial polycrystalline Cu foils. Monolayer graphene is abbreviated as MLG. **d** Relationship between the graphene bilayer coverage and the flow rate of $CO_2$, together with a linear fitting (dashed red). Inset: Typical OM images of the graphene grown using 5 sccm (top left) and 20 sccm (bottom right) $CO_2$. **e** Relationship between the graphene growth time and its bilayer coverage when 30 sccm $CO_2$ was introduced to grow graphene for 20 min, together with a linear fitting (dashed red). **f** Relationship between the graphene growth time and its bilayer coverage when 30 sccm $CO_2$ was utilized only

in the first 10 min. The dashed line connects the points using a B-spline option in Origin. Inset: OM image of the synthesized graphene after flowing $H_2$ and $CH_4$ for 90 min but only flowing $CO_2$ in the first 10 min. Error bars in (**d**−**f**) represent standard deviations from three measurement results for each sample. **g** Relationship between the graphene growth time and its bilayer coverage, showing the advantage of our $CO_2$-assisted strategy (red) in comparison with previous works (blue). **h** Photographs of the large-area BLG films grown on eight pieces of commercial Cu foils in one batch. **i** Statistic of the graphene bilayer coverage of the graphene samples in (**h**) (red) and the graphene sample grown without $CO_2$ (green).

sample with ~50% bilayer coverage was used to easily distinguish the MLG and BLG regions, and the sample was synthesized by alternately introducing $^{13}CH_4$ and $^{12}CH_4$ every 2 min for 10 min. As indicated, the top-layer graphene is mainly composed of $^{12}C$, while the bottom-layer graphene is composed of ~50% $^{12}C$ and 50% $^{13}C$ (Fig. 2g and Supplementary Fig. 16), verifying that the second-layer graphene grows underneath the first-layer graphene, consistent with the Raman results of the $^{13}C/^{12}C$-BLG treated after mild etching using oxygen plasma (Supplementary Fig. 17). Interestingly, we observed a high spatial uniformity of $^{13}C$ signals with a lower intensity than $^{12}C$ in the first-layer graphene, which is mainly attributed to the etching by $CO_2$ and repairing by $^{13}C$-labeled methane of the first-layer graphene (Supplementary Figs. 18 and 19)[54], implying the crucial role of the partial etching of the first-layer graphene for the fast synthesis of BLG on Cu (Supplementary Fig. 20).

Therefore we propose that the $CO_2$-assisted growth of BLG with full coverage on Cu substrate mainly consists of four steps: (1) a continuous MLG film (first-layer graphene) would quickly form on Cu

owing to the sufficient supply of carbon source, which can be etched with the formation of point defect by the introduction of $CO_2$; (2) active carbon species would diffuse through defects in first-layer graphene to fuel the nucleation and growth of second-layer graphene between the first-layer graphene and Cu substrate; (3) the growth of second-layer graphene domains are continuously fueled by the supply of active carbon species; (4) a continuous BLG film quickly forms on the Cu foil.

## Characterization of BLG grown on polycrystalline Cu foils

Stacking order and crystallinity of the as-synthesized BLG on the commercial polycrystalline Cu foils was characterized using TEM. Figure 3a displays a typical high-resolution TEM (HRTEM) image of BLG, from which the lattice of AB-BLG is clearly visible. Layer-by-layer etching of the as-synthesized graphene film using electron beam radiation also provides direct evidence about the graphene layer number. Selected area electron diffraction (SAED) patterns (inset in Fig. 3a) were collected across the whole region of the 3-mm-sized TEM

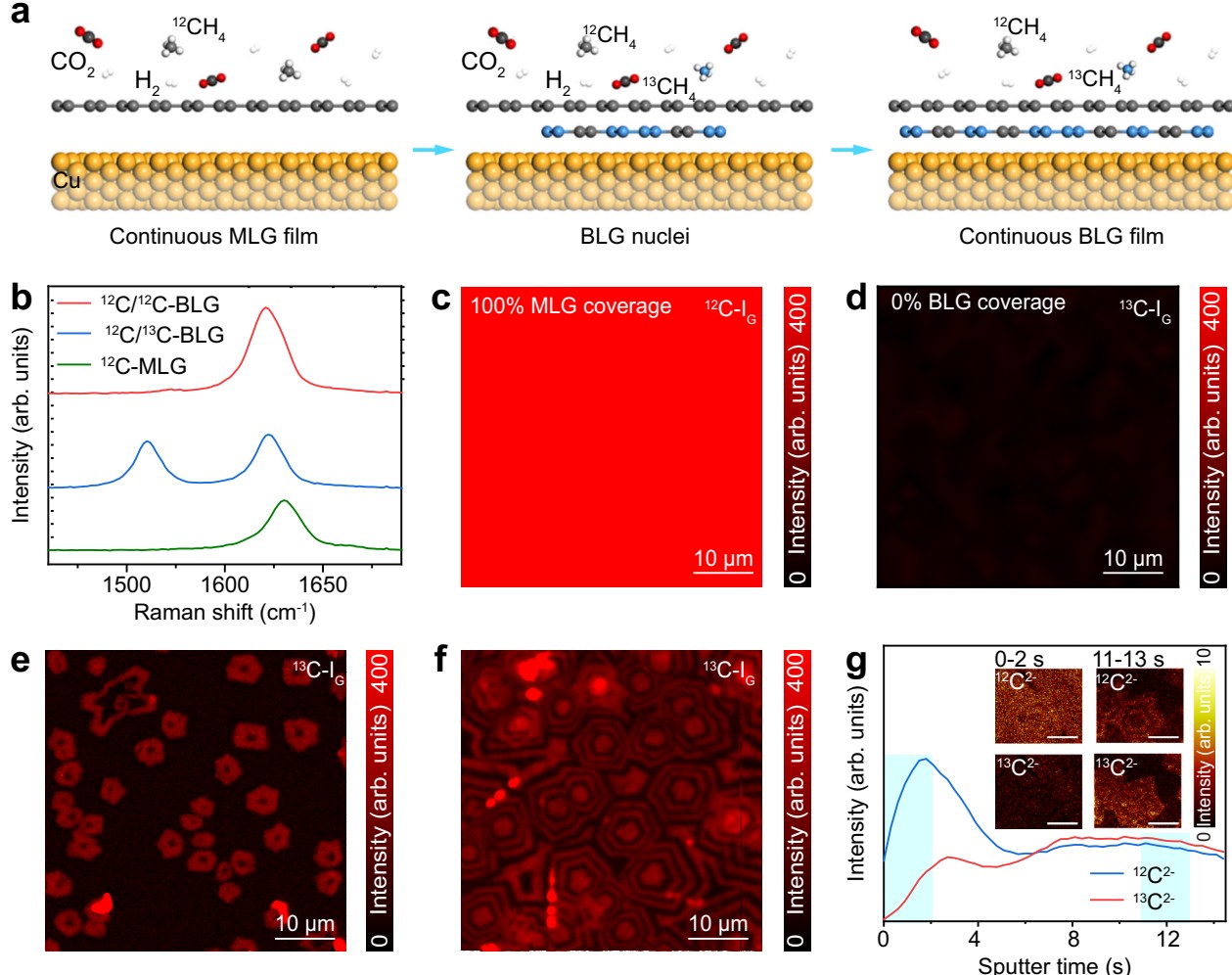

**Fig. 2 | Growth dynamic of the bilayer graphene (BLG) revealed by the isotope labeling technique. a** Schematic of the $CO_2$-assisted BLG growth via the alternative injection of $^{12}CH_4$ and $^{13}CH_4$. **b** Raman spectra of the $^{12}$C-monolayer graphene (MLG) (green), $^{12}$C/$^{12}$C-BLG (red), and $^{12}$C/$^{13}$C-BLG (blue). **c, d** Raman intensity mappings of $^{12}$C-G band ($^{12}$C-$I_G$) (**c**) and $^{13}$C-G band ($^{13}$C-$I_G$) (**d**) of the graphene whose growth time is 1 min. **e** Raman intensity mapping of $^{13}$C-G band ($^{13}$C-$I_G$) of the graphene whose growth time is 10 min. **f** Raman intensity mapping of the $^{13}$C-G band ($^{13}$C-$I_G$) of the graphene whose growth time is 20 min. **g** Depth profiles for $^{12}C^{2-}$ (blue) and $^{13}C^{2-}$ (red) of the BLG with ~50% bilayer coverage. Inset: Spatial distribution of $^{12}C^{2-}$ (top) and $^{13}C^{2-}$ (bottom) in the scanned regions during the sputter time of 0–2 s (left) and 11–13 s (right). The scale bar in the inset is 10 μm.

grid (inset in Fig. 3b). Based on orientations of the SAED patterns, the ratio of AB-BLG and tBLG was calculated. Note that, for AB-BLG, the hexagonal SAED pattern with a diffraction intensity ratio of the outer {1-210} peak over the inner {0-110} peak is 2:1, while for tBLG, there are two groups of MLG SAED patterns, whose intersection angle corresponds to the twist angle of tBLG (Supplementary Fig. 21). AB-BLG stacking structure was dominant on Cu(100)-dominated polycrystalline Cu foil substrates, with the ratio as high as 61% (Fig. 3b), presumably owing to the lower formation energy of AB-BLG than that of tBLG[55].

To further characterize the crystallinity of the BLG, large-area HRTEM images of AB-BLG and tBLG with lattice resolution were also captured (Supplementary Fig. 22), and the absence of defects clearly confirms the high quality of the BLG. The mechanical property of the suspended BLG, which is highly related to its defect density, was also measured using the atomic force microscopy (AFM) nano-indentation method[16]. From the force-displacement curve in Fig. 3c, Young's modulus and fracture force of the as-prepared BLG were estimated to be ~698 N m⁻¹ (1.04 TPa) and ~79.5 N m⁻¹ (118.6 GPa), which are comparable with the values of the exfoliated BLG and significantly higher than those of its monolayer counterpart (Supplementary Fig. 23).

The optical and electrical properties of the BLG films were investigated by transferring graphene films onto the functional substrates with the assistance of polymethyl methacrylate (PMMA)[56]. As shown in Fig. 3d, the BLG film on the quartz substrate exhibits an average optical transmittance of ~95.4% at 550 nm wavelength with high uniformity in large area (Supplementary Fig. 24). At the same time, the sheet resistance of BLG (~150 Ω sq⁻¹) on SiO₂/Si substrate is much lower than that of the MLG (~339 Ω sq⁻¹) (Fig. 3e and Supplementary Fig. 25). Furthermore, the narrow distribution of the sheet resistance of the BLG over large area further confirms its high uniformity (Fig. 3f).

### Synthesis of AB-BLG on Cu(111) substrates
Based on our $CO_2$-assisted strategy, two kinds of Cu(111) substrates were prepared as substrates for growing single-crystal AB-BLG. The ratio of AB-BLG increased to ~96% when using a single-crystal Cu(111) foil substrate derived from the high-temperature annealing of commercial polycrystalline Cu foils[57] (Supplementary Figs. 26 and 27). Moreover, a 100% AB-stacking structure has been successfully synthesized on the ultraflat single-crystal Cu(111) that was obtained by epitaxial growth of Cu(111) (i.e., Cu(111) film) on annealed c-plane

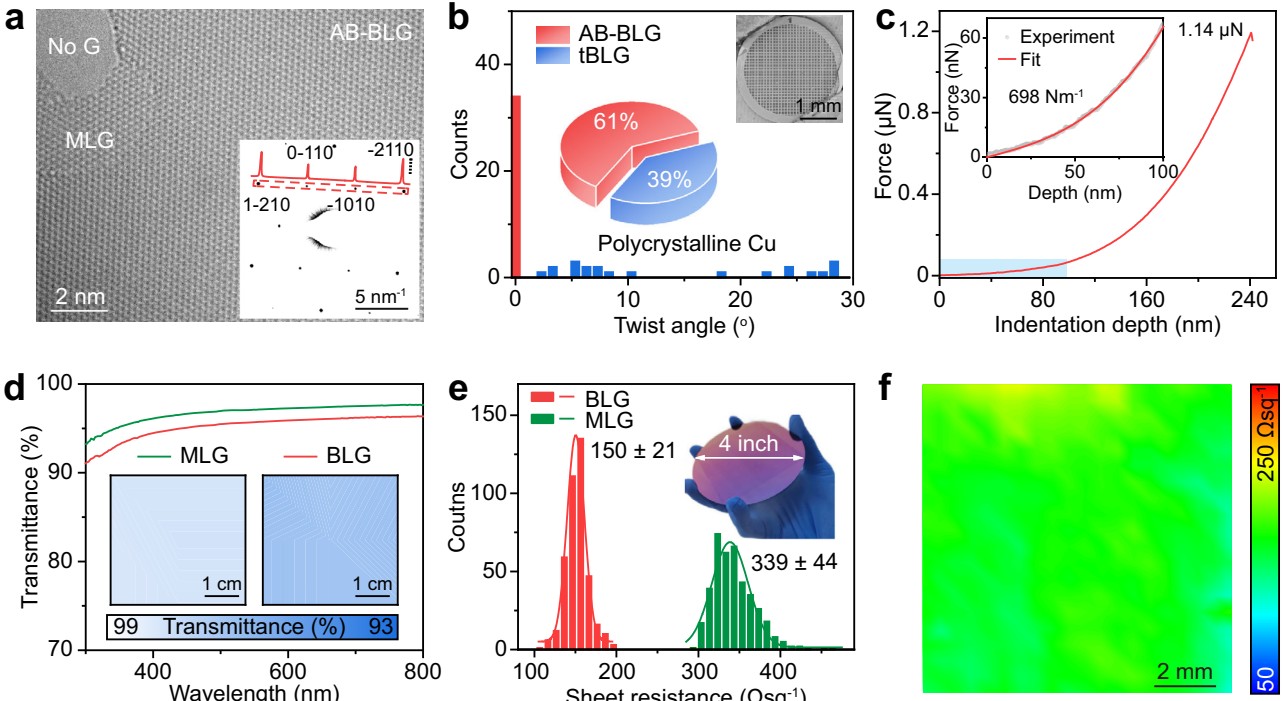

Fig. 3 | **Characterization of the continuous bilayer graphene (BLG) grown on polycrystalline Cu foils. a** High-resolution transmission electron microscope (HRTEM) image of the AB stacking-BLG (AB-BLG) with lattice resolution. Inset: Selected area electron diffraction (SAED) pattern of the AB-BLG and the intensity profile along the red line. **b** Distribution of twist angles based on SAED patterns of the BLG grown on the Cu(100)-dominated polycrystalline Cu. Inset: Statistical results of the distribution of AB-BLG (red) and non-AB stacking BLG (tBLG, blue) (center) and low-magnification scanning electron microscope image of the BLG transferred onto a transmission electron microscope grid (top right). **c** Atomic

force microscope nano-indentation measurement result of the BLG, for which the shaded area is used to fit the Young modulus of the BLG. Inset: Magnified image of the shaded area. **d** UV-vis spectra of the monolayer graphene (MLG, green) and BLG (red) transferred onto quartz substrates. Inset: Large-area transmittance mappings of the MLG (left) and BLG (right) at 550 nm wavelength. **e** Statistical histograms of the sheet resistance of the MLG (green) and BLG (red) transferred onto $SiO_2$/Si substrates. Error bars represent standard deviations from the mapping results. Inset: photograph of the large-area BLG transferred onto a 4-inch-sized $SiO_2$/Si substrate. **f** Sheet resistance mapping result of the BLG.

sapphire[58] (Fig. 4a and Supplementary Figs. 28–30), which might be thanks to the enhanced surface flatness and purity of the Cu(111) film substrate. The atomic-resolution scanning TEM (STEM) image of AB-BLG was also acquired, in which the bright and dark spots correspond to the overlapped AB carbon atoms and the un-overlapped A or B carbon atoms, respectively (inset in Fig. 4a).

Raman spectra of the AB-BLG films grown on Cu(111) were collected after the transfer of graphene to $SiO_2$/Si substrate (Supplementary Figs. 31 and 32), based on which the stacking order and defect density of as-received graphene can be inferred[16]. The absence of the D band further verifies the high crystallinity of the BLG (Fig. 4b). The asymmetrical 2D band with a full width at half maximum (FWHM) of ~55 cm$^{-1}$ can be well fitted into four Lorentzian bands with different frequencies, confirming the AB stacking order (Fig. 4c). In addition, Raman mapping and statistics of both the FWHM of 2D band and intensity ratio of 2D band to G band (Supplementary Fig. 31b, c) further reveal the high quality and spatial uniformity of the AB-BLG.

The electrical quality of the AB-BLG is probed using the dual-gate Hall bar device, where the AB-BLG crystal is encapsulated by two bulk hBN flakes (Supplementary Figs. 33 and 34)[59], which function as top and bottom dielectrics. An observable bandgap can be induced in our AB-BLG device by applying an out-of-plane electric displacement field[13,14]. In detail, the total resistance ($\rho_{xx}$) of the BLG varied significantly with both back gate voltage ($V_{bg}$) and top gate voltage ($V_{tg}$), which clearly verified the tunability of the bandgap (Fig. 4d and Supplementary Fig. 35).

Compared with MLG, BLG also shows superior performance when utilized for graphene-based photodetection. The adjacent MLG and AB-BLG regions, after transferring onto a $SiO_2$/Si substrate,

were etched into a strip and then fabricated into a two-terminal field effect transistor (FET) device (Inset of Fig. 4e and Supplementary Fig. 36). After conducting the photocurrent mapping over the entire device using a home-built scanning photocurrent microscopy, a strong photoresponse was observed at the interfaces of the graphene-metal electrodes with opposite polarity (Fig. 4f), consistent with previously reported results of metal-graphene junctions[60,61]. The photocurrent profiles along the two graphene/electrode interfaces are displayed in Fig. 4e, in which a much larger net photocurrent was generated in the BLG/metal junction than that between the MLG/metal junction, which can be attributed to the enhanced optical adsorption and higher Seebeck coefficient of BLG[62,63]. Enhanced photoresponsivity with uniform distribution is also observed in the channel region of the BLG in comparison with its monolayer counterpart (Fig. 4f).

## Discussion

In all, we raise a strategy to synthesize meter-sized BLG films on commercial polycrystalline Cu foils by introducing trace $CO_2$ during the high-temperature growth stage, based on which continuous BLG is obtained within 20 min. The BLG is dominated by AB-stacking structure (~61% areal ratio) and exhibits improved mechanical, optical, and electrical properties. Moreover, ~100% AB-BLG single crystal has been successfully grown on the ultraflat single-crystal Cu(111)/sapphire substrate, which is prepared by epitaxial deposition of Cu on single-crystal sapphire substrates. This work not only proposes an effective mechanism for the synthesis of BLG but also paves the way for the fast production and applications of large-scale BLG films.

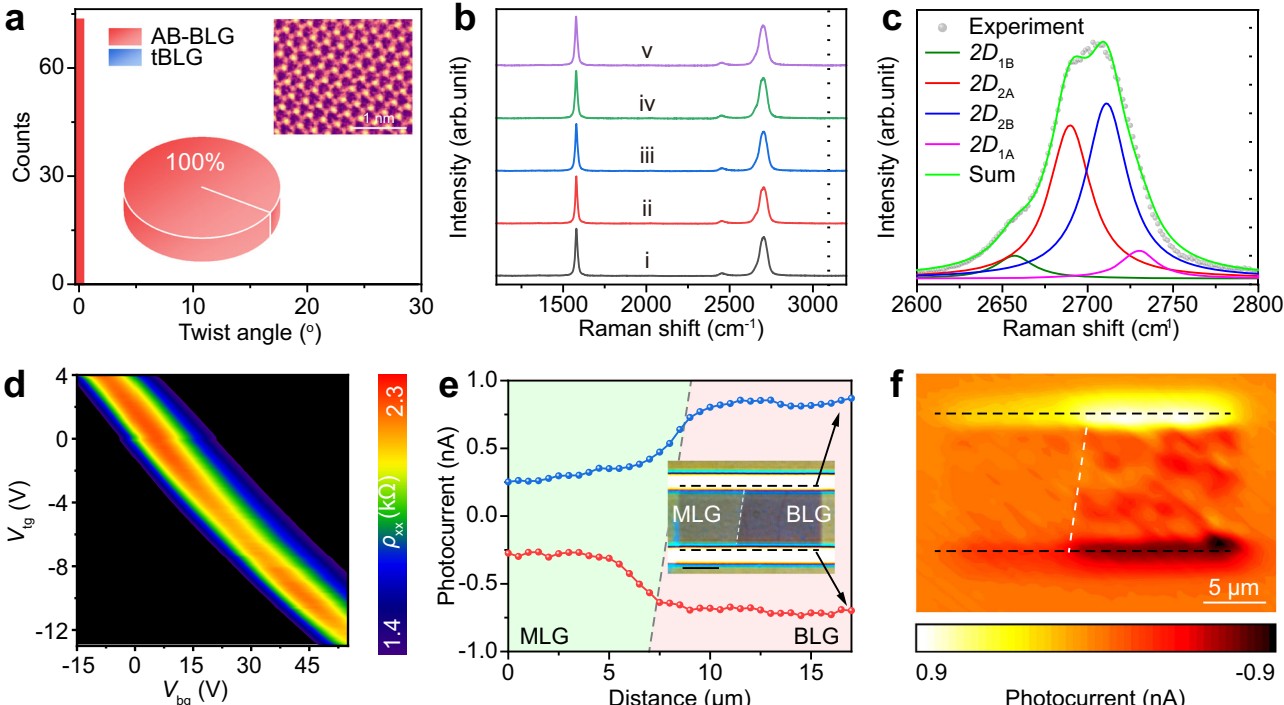

**Fig. 4 | Synthesis of AB-stacking bilayer graphene (AB-BLG) on Cu(111) substrates. a** Distribution of twist angles based on selected area electron diffraction patterns of the BLG grown on the ultraflat single-crystal Cu(111) substrate that was obtained by epitaxial growth of Cu(111) (i.e., Cu(111) film) on annealed c-plane sapphire. Inset: Statistical results of AB-BLG (red) and non-AB stacking BLG (tBLG) (blue) (center) and scanning transmission electron microscope image of the AB-BLG with atomic resolution (top right). **b** Raman spectra of the AB-BLG transferred onto a SiO$_2$/Si substrate acquired via a line scanning with the step of 50 μm by normalizing the G band intensity. **c** Raman 2D band of the AB-BLG, which is fitted by four Lorentzian peaks. **d** Two-dimensional plot of the total resistance ($\rho_{xx}$) as functions of both top gate voltage ($V_{tg}$) and back gate voltage ($V_{bg}$) of a dual-gate AB-BLG device. **e** Photocurrent distribution in the two graphene/electrode junctions. Inset: Optical microscope image of the graphene field effect transistor (FET) device and the boundary of monolayer graphene (MLG) and BLG regions is marked by the white dashed line. Scale bar: 5 μm. The green and purple areas correspond to the MLG and BLG regions, respectively, and the blue and red curves correspond to the two black dashed lines in the inset, as denoted by the black arrows. **f** Photocurrent mapping result of the graphene device, in which the MLG is on the left part and the BLG is on the right part, where the two black dashed lines in (**f**) correspond to the regions marked by black dashed lines in the inset of (**e**) and the white dashed line denotes the boundary of MLG and BLG regions.

## Methods

### Graphene synthesis

The graphene film is grown on commercial Cu foils (25 μm thick, 99.9%, Kunshan luzhifa Electronic Technology Co., Ltd) in a hot-wall low-pressure CVD system equipped with a quartz tube (6 inches in diameter). The Cu foil is first heated to 1020 °C under CO$_2$ atmosphere (500 sccm, ~1000 Pa) for 60 min and then annealed in the same atmosphere for 30 min to eliminate the carbon-containing contamination before graphene growth. A mixture gas of H$_2$ (100–1000 sccm), CH$_4$ (1–20 sccm), and CO$_2$ (0–30 sccm) is subsequently flowed into the CVD chamber to initiate the BLG growth (1–90 min). CO$_2$ is turned off after the graphene growth, followed by rapid cooling to room temperature under the mixing gas of CH$_4$ and H$_2$. In addition, to investigate the growth mechanism, the $^{13}$C-labeled CH$_4$ is utilized, which is purchased from the Sigma-Aldrich company (production number #490229) with $^{13}$C atom ratio of 99%. The batch-to-batch mass production of eight pieces of submeter-sized BLG films (0.3 × 0.1 m$^2$) with the aid of CO$_2$ is conducted by supporting the Cu foils using quartz plates, which are stacked vertically using the small quartz columns as the spacer to control the distance between adjacent layers. The CO$_2$ annealing time is prolonged to 60 min to thoroughly clean the Cu surface while other parameters were kept the same. Meter-sized BLG film is grown inside a quartz tube with a diameter of 10 inches using a homemade roll-to-roll CVD system, whose constant temperature region is 2 m long (Details seen in Supplementary information). For single-crystal AB-BLG grown on Cu(111) films, after 30 min high-temperature annealing at 1020 °C using 1000 sccm Ar and 50 sccm H$_2$, 200 sccm CH$_4$ (0.1%, diluted in Ar) is introduced for 60 min to acquire a continuous MLG film, followed by flowing 30 sccm CO$_2$ for 30 min to prepare a continuous BLG.

### Graphene transfer

The graphene is transferred to quartz and SiO$_2$/Si substrates with the assistance of PMMA for structure characterization and property measurement[64]. In brief, PMMA is spin-coated atop the graphene/Cu samples and then baked at 170 °C for 5 min. Prior to Cu etching using 1 M Na$_2$S$_2$O$_8$ solution, the graphene film on the other side of Cu is removed using air plasma (40 W, 15 sccm, 3 min) (Pico SLS, Diener). After being washed several times using deionized water, the PMMA/graphene membrane is lifted by target substrates and dried overnight, followed by the PMMA dissolution in acetone. For TEM characterization, BLG films are transferred from the Cu substrates onto the commercial TEM grids (Quantifoil, Au-300 mesh-R2/1 μm) without using PMMA[65]. After putting a TEM grid on top of the flat BLG/Cu sample, a droplet of isopropanol was used to adhere them together, followed by etching Cu using Na$_2$S$_2$O$_8$ solution (0.5 M concentration) and cleaning the suspended BLG sample in distilled water, which is finally dried using a mild N$_2$ flow.

### Graphene characterization

Bilayer coverage of graphene films is evaluated using OM (Nikon microscopy (LV100 ND) and scanning electron microscopy (SEM, FEI Quattro S, acceleration voltage 1–20 kV). The crystallographic orientation of Cu is confirmed via EBSD (DigView 5 operated at 20 kV

voltage). The distribution of $^{12}$C and $^{13}$C is analyzed with a ToF-SIMS 5 instrument (ION-ToF GmbH, Münster, Germany), which is equipped with a 30 keV Bi$^{3+}$ primary ion gun, a 1 keV Cs$^{+}$ sputter gun, together with an electron flood gun for charge neutralization. TEM characterizations are conducted using FEI Tecnai F30 for collecting SAED patterns under 300 kV, using an aberration-corrected and monochromated G$^2$ cubed Titan 60-300 electron microscope for HRTEM imaging under 80 kV, and using a double aberration-corrected FEI (Titan Cubed Themis G2 electron microscope) for STEM imaging under 60 kV. The nano-indentation experiment is performed using Asylum Cypher ES AFM. Raman spectra were obtained with LabRAM HR-800 using a 532 nm laser and ×100 objective. Optical transmittance spectra are collected using a Perkin-Elmer Lambda 950 UV-vis spectrophotometer, and the transmittance mapping is conducted by manually adjusting the sample positions. Sheet resistance is measured by a four-probe resistance measuring meter (CDE ResMap 178).

### Device fabrication and measurement

For the electrical measurement, the AB-BLG is first encapsulated by two pieces of relatively thick hBN crystals (~40 nm) using the dry-peel technique with the aid of poly-propylene carbonate/poly dimethyl siloxane stack[59,66] and then fabricated into a Hall bar device with one-dimensional contacts (Cr/Au, 3 nm/80 nm) using electron-beam lithography (EBL) and standard etching procedure. The fabricated device is then measured using the conventional lock-in technique. The carrier mobility was measured at 300 K in a glovebox filled with Ar, and the gate-dependent transfer curves were measured at 290 K under vacuum after storage in high vacuum (~10$^{-7}$ torr) for 1 week. For the photocurrent measurement, the AB-BLG FET device is fabricated on a 285 nm SiO$_2$/Si substrate. The metal contacts (Ti/Au, 5 nm/45 nm) are fabricated using EBL, followed by metal deposition and the lift-off process. The photocurrent is then measured using a homemade scanning photocurrent microscope[67] under a focused 532 nm laser spot (~10 µW, 1 µm in diameter) without applying source-drain bias and gate bias.

## Data availability

The source data underlying the figures of this study are available at https://doi.org/10.6084/m9.figshare.22633576. All raw data generated during the current study are available from the corresponding authors upon request.

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

## Acknowledgements

This work was financially supported by the National Natural Science Foundation of China (Nos. T2188101 (Z.L.), 52072042 (J.Z.), 21525310 (H.P.), 11904389 (Z.L.), 21927804 (F.W.), and 22004121 (F.W.)), the National Basic Research Program of China (No. 2018YFA0703502 (Z.L.)), and Beijing National Laboratory for Molecular Sciences (BNLMS-CXTD-202001 (Z.L.)). The authors acknowledge Molecular Materials and Nanofabrication Laboratory (MMNL) in the College of Chemistry and Peking Nanofab at Peking University for the use of instruments.

## Author contributions

Z.L., L.L. H.P., and J.Z. conceived the experiment. Z.L., L.L., H.P., and H.L. supervised the project. J.Z., X.L., M.Z., and X.G. conducted the growth of BLG. X.L., X.Z., Y.L., Q.Y., L.S., J.Sh., and Q.L. conducted the mass production of BLG. J.Z., X.L., and K.J. conducted the clean transfer of graphene. X.S. and W.Y. conducted the first-principle simulations. X.L., M.Z., and X.Z. conducted the polymer-assisted transfer of BLG. J.Z., X.L., M.Z., X.Z., Z.Y., J.Sh., Y.L., X.G., H.C., and L.Z. took and analyzed the OM, SEM, UV-vis, AFM, Raman and sheet resistance data. T.F. and F.W. performed ToF-SIMS. X.L., H.T., M.R., T.G., and F.L. conducted the TEM experiment and data analysis. W.Z. and X.W. measured the mechanical property. W.W. and R.Z. fabricated and measured the BLG Hall bar device. J.Su. and J.Y. fabricated and measured the photodetector. All authors discussed the results and wrote the manuscript.

## Competing interests

The authors declare no competing interests.

## Additional information

[1]Center for Nanochemistry, Beijing Science and Engineering Center for Nanocarbons, Beijing National Laboratory for Molecular Science, College of Chemistry and Molecular Engineering, Peking University, 100871 Beijing, P. R. China. [2]Beijing Graphene Institute, 100095 Beijing, P. R. China. [3]Academy for Advanced Interdisciplinary Studies, Peking University, 100871 Beijing, P. R. China. [4]Department of Engineering, University of Cambridge, Cambridge CB3 0FA, UK. [5]School of Material Science and Engineering, Tianjin Key Laboratory of Advanced Fibers and Energy Storage, State Key Laboratory of Separation Membranes and Membrane Processes, Tiangong University, 300387 Tianjin, P. R. China. [6]Department of Physics and Astronomy, University of Manchester, Manchester M13 9PL, UK. [7]Leibniz Institute for Solid State and Materials Research Dresden, P.O. Box 270116, D-01171 Dresden, Germany. [8]State Key Laboratory for Turbulence and Complex System, Department of Mechanics and Engineering Science, College of Engineering, Peking University, 100871 Beijing, P. R. China. [9]CAS Key Laboratory of Analytical Chemistry for Living Biosystems, Institute of Chemistry, Chinese Academy of Sciences, 100190 Beijing, P. R. China. [10]Beijing National Laboratory for Molecular Sciences, National Centre for Mass Spectrometry in Beijing, CAS Key Laboratory of Analytical Chemistry for Living Biosystems, Institute of Chemistry, Chinese Academy of Sciences, 100190 Beijing, P. R. China. [11]Soochow Institute for Energy and Materials Innovations, Soochow University, 215006 Suzhou, P. R. China. [12]Centre of Polymer and Carbon Materials, Polish Academy of Sciences, M. Curie-Skłodowskiej 34, Zabrze 41-819, Poland. [13]Institute of Environmental Technology, VŠB -Technical University of Ostrava, 17 Listopadu 15, Ostrava 708 33, Czech Republic. [14]School of Materials Science and Engineering, Peking University, 100871 Beijing, P. R. China. [15]These authors contributed equally: Jincan Zhang, Xiaoting Liu, Mengqi Zhang, Rui Zhang. [16]These authors jointly supervised this work: Haihui Liu, Hailin Peng, Li Lin, Zhongfan Liu. ✉e-mail: liuhaihui@tiangong.edu.cn; hlpeng@pku.edu.cn; linli-cnc@pku.edu.cn; zfliu@pku.edu.cn

