## [Peer Review File · Nature Communications]

Fast synthesis of large-area bilayer graphene film on CuREVIEWER COMMENTS

Reviewer #1 (Remarks to the Author):

Zhang et. al. developed a method to synthesize a meter-sized bilayer graphene (BLG) film on Cu substrate by introducing trace CO₂ during the high-temperature growth process within only 20 min. Multiple characterization tools are performed to analyze the number of layers, growth mechanism, mechanical and electrical properties of their graphene film. The work appears to be carried out with care and the results promise a step forward by providing a new route toward BLG across large areas. BLG growth with assistance of CO₂ is new and interesting. However, the manuscript has a number of weaknesses and the conclusion was not fully supported by appropriate data. Therefore, I cannot recommend the publication of this work at this stage. The following comments/concerns have to be fully addressed before publication.

Major:

1. As emphasized in the title and abstract, this work provide a method to synthesize large-scale BLG up to meter-scale. But I do not see any data to prove this main claim in the entire manuscript. Only the photograph of Cu foil in meter size was provided. Fig 1c shows only 100 x 100 μm^2 BLG. I suggest to take optical microscope (OM) continuously across 1 mm (at least). Or LEED pattern of BLG/Cu across larger area could help.
2. It's very confusing in the abstract and introduction to distinguish two types of graphene (growth on Cu foil and Cu film). Authors did not mention tBLG in the abstract and introduction. So, my first impression is that authors are able to obtain meter-sized 100% AB-BLG. Authors argue in Fig. 3b,c that they are able obtain 39% tBLG + 61%AB-BLG (on Cu foil, meter-scale) and 100% AB-BLG (on Cu film, scale was not mentioned). Please classify this.
3. In Fig 4, what kind of BLG authors used to measure sheet resistance (4b) and dual-gate device (4d)?. If author used AB-BLG, why sheet is reduced for BLG in comparison with MLG (b)?. Authors claim in the main text that they observed the tunability of the band gap, so BLG for this device should be AB-BLG. But I did not see the resistance varied with both V_{bg} and V_{tg} (Fig. 4d). Could author provide I-V curve of Fig. 4d.
4. To create the point defect for second layer growth, authors employed CO₂ (in the main body of manuscript) and plasma etching (Supplementary Note 2). It seems like, once point defect was created, second layer could be grown underneath first layer. H₂ also can etch graphene to create point defect, Could author attempt to create it by H₂ and grow second layer. I am wondering the role of CO₂, may be it is not only for etching to create defect.
5. In previous work (Nano Lett. 2010, 10, 11, 4702-4707), BLG also could be grown in wafer-scale within few tens of min on Cu substrate. Could authors clarify the advantage of their work compare to this previous work?.
6. Is tri-layer or tetra-layer graphene possible using this CO₂ assisted growth?. By simple thinking, after finishing 2nd layer growth. CO₂ could be introduce again to etch graphene, leaving space for carbon species diffusing into Cu for tri-layer or tetra-layer graphene growth.
7. To prove second layer grow underneath first layer, author performed the depth profile mapping (Fig. 2c). After 11-13s sputtering, there is ring-like pattern in 12C2-. Why this ring-like pattern did not appear in 13C2-. And in Supplementary Note 3, authors performed mild oxygen plasma etching to etch top layer graphene without destroying the bottom layer. Could author provide optical microscopy image of BLG before and after mild plasma etching to prove layer by layer etching?

Minor

1. Fig. 1c, author should point out reference number directly in the panel. It's very difficult to follow up if all reported works was marked as blue circle.
2. Line 109: Supplementary Fig. 4 was wrongly mentioned
3. To much data in the main Figure. Please make it concisely by moving not important data to Supporting information.

Reviewer #2 (Remarks to the Author):

Authors report the fast synthesis of large-area bilayer graphene (BLG) on Cu by introducing trace amount of CO₂ during a high-temperature growth process. Bilayer graphene exhibits tunable twist

angle and band structure with high mechanical strength, low resistance, and uniform transmittance. These unique properties hold immense promise for next-generation devices. The manuscript presents a great opportunity to reduce CO₂ and methane into a useful BLG growth for fundamental physics studies. However, there are some challenges with growth mechanism of this work, hence I recommend that the manuscript is published in Nature Communication after minor revision.

I left some comments to enhance the quality of this manuscript.

Comments

1. First, a fast synthesis of AB-BLG with the assistance of CO₂ reduction has been reported in Gong et al ACS Cent. Sci. 2022, 8, 394–401. They proposed that CO₂ can simultaneously serve as an etching gas and carbon source for graphene growth. Oxygen atoms provided by CO₂ create defects on monolayer graphene (MLG), which break the MLG self-limiting growth and host the nucleation sites for BLG. Could authors distinguish their work from this work? Does carbon in CO₂ play a role in the graphene formation as reported by Gong et al ACS Cent. Sci. 2022, 8, 394–401 and Seekwaew et al Membranes 2022, 12, 796? Also, did authors confirm the growth of BLG using only trace CO₂ supply during their growth process? Authors should clearly discuss the research trend of CO₂ assistance for BLG growth to enhance the novelty of this work.

2. Apart from the penetrative mechanism of second layer graphene as suggested by J. Phys. Chem. C 2014, 118, 6201–6206, is there the possibility of defects generation and formation of second growth on top of the first layer during growth process?

3. On page 8, lines 168-17: authors stated, "Note that, for AB-BLG, the hexagonal SAED pattern with diffraction intensity ratio of the outer {1-210} peak over the inner {1-110} peak is 2:1 (Fig. 3a)". However, in Fig. 3a, labeling reads (1-210 and 0-110). Authors should clarify this.

Reviewer #3 (Remarks to the Author):

This paper "Fast synthesis of large-area bilayer graphene film on Cu" by J. Zheng et al., reports the controlled synthesis of AB-stacked BLG by introducing CO₂ gas in CH₄ feedstock. The authors claim that they can grow uniform BLG in short time (20 min). In addition, centimeter or meter-scale BLG can be also synthesized by large Cu foils. It seems that CO₂ acts as weak oxidant that makes holes to grow the second graphene layer underneath the monolayer graphene. Their finding is interesting and important. However, many papers have been already published about the growth of uniform, AB-stacked BLG by CVD. Although the finding of CO₂-assisted BLG is interesting, I feel that the findings are not new enough to be published in Nature Communications. In addition, as listed below, there are many weaknesses in this manuscript. Therefore, I suggest publication in more specialized journals.

1) There are many papers reporting the CVD growth of bilayer, monolayer, and few-layer graphene using CO₂ gas. However, in this manuscript, the authors cited none of these papers, which is not fair.

- P. Gong et al., Precise CO₂ Reduction for Bilayer Graphene, ACS Central Science 8, 394, (2022)

- Y. Seekaew et al., Conversion of Carbon Dioxide into Chemical Vapor Deposited Graphene with Controllable Number of Layers via Hydrogen Plasma Pre-Treatment, Membranes, 12, 796 (2022)

- A. J. Strudwick et al., Chemical Vapor Deposition of High Quality Graphene Films from Carbon Dioxide Atmospheres, ACS Nano 2015, 9, 1, 31 (2015).

- A. K. Grebenko et al., High-Quality Graphene Using Boudouard Reaction, Adv. Sci. 9, 2200217 (2022).

- M. K. Kairi et al., Co-synthesis of large-area graphene and syngas via CVD method from greenhouse gases, Mater. Lett. 227, 132 (2018).

2) The authors' claim of "100% AB" in the abstract is not solid. They concluded "100% AB" by measuring 100 electron diffraction patterns (Fig. 3c), but it is not clearly written how and where they measured the diffraction patterns. If they measured one TEM specimen (3 mm) or some of holes in one grid, the conclusion is not so reliable. More careful investigation combining other methods, such as Raman, as well as detailed explanation of the method of the AB ratio

determination is required.

More importantly, in Fig. 1b, not all the hexagonal BLG grains are not aligned. It is not clear whether the authors used Cu(100) foil or Cu(111) foil in Fig. 1b, but this is strange for me. If Fig. 1b is transferred from a Cu(100) foil, the authors should present optical images of aligned BLG grains transferred from a Cu(111) foil or a Cu(111) film as well.

3) Can the author make AB 100% BLG in 20 min? It is unclear 20 min is time for making a full BLG sheet or for growing AB 100% BLG. The authors' claim is not very clear.

4) Line 85: The authors claim "meter-sized BLG films high uniformity". However, when I carefully check Fig. 1c, there are at least two 3L graphene grains at the left bottom corner and right edge of Fig. 1c. This indicates a possibility that there can be a number of 3L grains in the BLG film, in contrast to the authors' claim ("meter-sized BLG films high uniformity").

5) If CO₂ makes holes for the entrance of carbon feedstock, there can be many defects in top graphene layer. This may result in a strong Raman D band. However, the Raman spectra in Fig. 3g shows negligible D band. Please explain this.

The followings are more minor comments:

6) Fig. 1i: No experimental evidence is shown. OM images for every 1cm distance should be presented to prove Fig. 1i.

7) Fig. 3g: All five Raman spectra looks quite similar. This makes me feel very strange, because there should be some fluctuations in 2D band shape and intensity. Please indicate where you measured each Raman by using optical images.

8) Fig. 4a inset: Please describe how you measured the transmittance mapping. Because a standard UV-vis spectrometer measures only at one specific point.

9) Fig. 4b: The sheet resistance is much lower than that widely reported for BLG (300-1000 Ohm/sq). Please describe how you measured the sheet resistance including the electrode size and spacing. Did the authors experience the BLG film breakage using CDE RsMap 178? This is because this equipment probably uses four pins directly attaching to a graphene sheet.

10) Fig. 4d: There is no transfer curves for the hBN/BLG/hBN device with different top and bottom gate voltages. Generally, researchers present transfer curves when they discuss the band gap opening in BLG samples. Also, the temperature of the device measurement should be provided.

11) Supplementary Fig. 14: It is unclear for me where and how you measured "Representative HRTEM images". More detailed explanation is necessary. Ideally, these images should be collected at different positions far apart each other.

12) Supplementary Fig. 17: Generally, the carrier mobility of BLG is low, due to its parabolic band structure. However, the authors observed very high carrier mobility over 5000 cm²/Vs. Why did they obtain such high mobility for AB BLG?

13) Fig. 1g: I think the references correspond to Supplementary Table 1. It is better for general readers to mention that the original references are cited in Supplementary Table 1.

Reviewer #1 (Remarks to the Author)

Zhang et. al. developed a method to synthesize a meter-sized bilayer graphene (BLG) film on Cu substrate by introducing trace CO₂ during the high-temperature growth process within only 20 min. Multiple characterization tools are performed to analyze the number of layers, growth mechanism, mechanical and electrical properties of their graphene film.

The work appears to be carried out with care and the results promise a step forward by providing a new route toward BLG across large areas. BLG growth with assistance of CO₂ is new and interesting. However, the manuscript has a number of weaknesses and the conclusion was not fully supported by appropriate data. Therefore, I cannot recommend the publication of this work at this stage. The following comments/concerns have to be fully addressed before publication.

Reply: We appreciate very much for the positive evaluation on the significance and novelty of our manuscript from the reviewer #1 and the constructive comments to strengthen our work for publication. Following the valuable suggestions from the reviewer #1, we have conducted additional experiments and theoretical calculations, and made point-to-point response and revisions as follows.

Major:

1. As emphasized in the title and abstract, this work provides a method to synthesize large-scale BLG up to meter-scale. But I do not see any data to prove this main claim in the entire manuscript. Only the photograph of Cu foil in meter size was provided. Fig 1c shows only 100 x 100 μm² BLG. I suggest to take optical microscope (OM) continuously across 1 mm (at least). Or LEED pattern of BLG/Cu across larger area could help.

Reply: Thank reviewer #1 for the valuable comments. Following the reviewer's suggestions, optical microscope (OM) images of the bilayer graphene (BLG) film continuously across 1 mm have been taken (Fig. R1) after being transferred to the SiO₂/Si substrate, which proves the successful synthesis of large-area BLG film.

Figure R1. OM images of the BLG film transferred on a SiO_2/Si substrate. **a-d** OM images of the BLG film acquired across 1 mm-sized region.

Meanwhile, low energy electron diffraction (LEED) patterns of the BLG grown on the $\text{Cu}(111)/\text{sapphire}$ substrate across cm-sized regions have also been acquired (Fig. R2). The single set of hexagonal diffraction spots with no rotational misalignment suggests the epitaxial growth of AB-stacking BLG (AB-BLG) on the single-crystal $\text{Cu}(111)$ substrate. The identical orientations of the LEED patterns acquired at three different positions across the whole sample ($2 \times 2 \text{ cm}^2$) further demonstrate the successful synthesis of the large-scale AB-BLG single-crystal film.

Figure R2. LEED patterns of the AB-BLG film grown on the $\text{Cu}(111)/\text{sapphire}$ substrate. **a**

Photograph of the cm-sized AB-BLG/Cu(111)/sapphire sample. **b-d** LEED patterns acquired on the AB-BLG/Cu(111)/sapphire sample.

Fig. R2 has been added as Supplementary Fig. 24 in the updated version to support our conclusion, together with relevant discussion, as below.

‘Moreover, low energy electron diffraction (LEED) patterns of the AB-BLG film grown on the Cu(111)/sapphire substrate were also acquired (Supplementary Fig. 24). The single set of hexagonal diffraction spots with no rotational misalignment suggests the epitaxial growth of AB-stacking BLG (AB-BLG) on the single-crystal Cu(111) substrate. The identical orientations of the LEED patterns acquired at three different positions across the whole sample (2 x 2 cm²) further demonstrate the successful synthesis of the large-scale AB-BLG single-crystal film in at least cm-sized regions.’ (Page 16, lines 11-17 in the updated supporting information)

2. It's very confusing in the abstract and introduction to distinguish two types of graphene (growth on Cu foil and Cu film). Authors did not mention tBLG in the abstract and introduction. So, my first impression is that authors are able to obtain meter-sized 100% AB-BLG. Authors argue in Fig. 3b,c that they are able to obtain 39% tBLG + 61% AB-BLG (on Cu foil, meter-scale) and 100% AB-BLG (on Cu film, scale was not mentioned). Please classify this.

Reply: Great thanks for the valuable comments. We apologize for the possible misleading. In this work, our primary focus is on the fast synthesis of large-area BLG on low-cost commercial Cu foils. By introducing CO₂ during the high-temperature growth process, the continuous BLG film is successfully synthesized on polycrystalline Cu foils within 20 min. This CO₂-assisted growth strategy is highly compatible with both batch-to-batch and roll-to-roll mass production processes and thus enables the scalable synthesis of BLG in meter size.

There are two kinds of single-crystal Cu substrates for growing BLG. The first one is the Cu(111) foil which derived from the high-temperature annealing (i.e. Cu(111) foil) of the low-cost commercial polycrystalline Cu foils (*ACS Nano* 2021, 16, 285) and the second one is the ultraflat single-crystal Cu(111) that was obtained by epitaxial growth of Cu(111) (i.e. Cu(111) film) on annealed *c*-plane sapphire (*ACS Nano* 2017, 11, 12337). In this regard, by using the Cu(111) foils and using the single-crystal Cu(111) prepared on the annealed *c*-plane sapphire,

the ratio of AB-BLG structures increases to 96% and 100%, respectively, both of which are much higher than the ratio grown on polycrystalline Cu foils ($\sim 61\%$).

To avoid misunderstanding, we have reorganized Figs. 3 and 4 in the main text to separate the experimental results of the BLG films according to the type of Cu substrates. Therefore, in the updated version, following Figs. 1 and 2, Fig. 3 only discusses the structure and property characterization of the large-area polycrystalline BLG film synthesized on commercial polycrystalline Cu foils (Fig. R3), while Fig. 4 focuses on the preparation and relevant characterizations of the AB-BLG grown on the Cu(111) substrates (Fig. R4).

Figure R3. Characterization of the continuous BLG grown on polycrystalline Cu foils. **a** HRTEM image of the AB-BLG with lattice resolution. Inset: SAED pattern of the AB-BLG and the intensity profile along the red line. **b** Distribution of twist angles based on SAED patterns of the BLG grown on the Cu(100)-dominated polycrystalline Cu. Inset: Statistical results of the stacking order (AB stacking or non-AB stacking) (centre) and low-magnification SEM image of the BLG transferred onto a TEM grid (top right). **c** AFM nano-indentation measurement result of the BLG. Inset: Fitting of the Young modulus. **d** UV-vis spectra of the MLG (green) and BLG (red) transferred onto quartz substrates. Inset: Large-area transmittance mappings of the MLG (left) and BLG (right) at 550 nm wavelength. **e** Statistical histograms of the sheet resistance of the MLG (green) and BLG (red) transferred onto SiO₂/Si substrates. Inset: Photograph of the large-area BLG transferred onto a 4 inch-sized SiO₂/Si substrate. **f** Sheet resistance mapping result of the BLG.

Figure R4. Synthesis of AB-BLG on Cu(111) substrates. **a** Distribution of twist angles based on SAED patterns of the BLG grown on the ultraflat single-crystal Cu(111) substrate that was obtained by epitaxial growth of Cu(111) (i.e. Cu(111) film) on annealed c-plane sapphire. Inset: Statistical results of the stacking order (AB stacking or non-AB stacking) (centre) and STEM image of the AB-BLG with atomic resolution (top right). **b** Raman spectra of the AB-BLG transferred onto a SiO₂/Si substrate acquired via a line scanning with the step of 50 μm by normalizing the G band intensity. **c** Raman 2D band of the AB-BLG, which is fitted by four Lorentzian peaks. **d** Two-dimensional plot of the ρ_{xx} as functions of both V_{tg} and V_{bg} of a dual-gate AB-BLG device. **e** Photocurrent distribution in the two graphene/electrode junctions. Inset: OM image of the graphene FET device. Scalebar: 5 μm. **f** Photocurrent mapping result of the graphene device, in which the MLG is on the left part and the BLG is on the right part.

The relevant discussions have also been updated accordingly, as below.

‘Bilayer graphene is intriguing for its unique properties and potential applications in electronics, photonics, and mechanics. However, the chemical vapor deposition synthesis of large-area high-quality bilayer graphene on Cu is suffering from low growth rate and limited bilayer coverage. Herein, we demonstrate the fast synthesis of meter-sized bilayer graphene film on commercial polycrystalline Cu foils by introducing trace CO₂ during the high-temperature growth. Continuous bilayer graphene with a high ratio of AB-stacking structure can be obtained within 20 min, which exhibits enhanced mechanical strength, uniform transmittance, and low sheet resistance in large area. Moreover, 96% and 100% AB-stacking structures were achieved in bilayer graphene grown on single-crystal Cu(111) foil and ultraflat

single-crystal Cu(111)/sapphire substrates, respectively. The AB-stacking bilayer graphene exhibits tunable bandgap and performs well in photodetection. This work provides a new insight both in the growth mechanism and in the mass production of large-area high-quality BLG on Cu. (Page 2, lines 40-52 in the new main text)

‘Transmission electron microscopy (TEM) and Raman measurements confirmed AB-stacking dominated in the as-received BLG, which also exhibits enhanced mechanical strength and reduced sheet resistance. Especially, 96% and 100% AB-stacking structure can be obtained on single-crystal Cu(111) foils and ultraflat single-crystal Cu(111)/sapphire substrates, respectively. Moreover, AB-BLG also exhibits tunable bandgap and promising performance as employed for photodetectors.’ (Page 5, Lines 104-109 in the new main text)

‘Characterization of the continuous BLG grown on polycrystalline Cu foils.’ (Page 8, line 185 in the new main text)

‘To further characterize the crystallinity of the BLG, large-area HRTEM images of AB-BLG and tBLG with lattice resolution were also captured (Supplementary Fig. 18), and the absence of defects clearly confirms the high quality of the BLG.’ (Page 9, lines 199-201 in the new main text)

‘Synthesis of AB-BLG on Cu(111) substrates.

Based on our CO₂-assisted strategy, two kinds of Cu(111) substrates were prepared as substrates for growing single-crystal AB-BLG. The ratio of AB-BLG increased to ~96% when using a single-crystal Cu(111) foil substrate which derived from the high-temperature annealing of commercial polycrystalline Cu foils⁵⁷ (Supplementary Fig. 21 and Fig. 22). Moreover, 100% AB-stacking structure has been successfully synthesized on the ultraflat single-crystal Cu(111) that was obtained by epitaxial growth of Cu(111) (i.e. Cu(111) film) on annealed c-plane sapphire⁵⁸ (Fig. 4a and Supplementary Figs. 23-25),’ (Page 10, lines 214-221 in the new main text)

‘Discussion

In all, we raise a new strategy to synthesize meter-sized BLG films on commercial polycrystalline Cu foils by introducing trace CO₂ during the high-temperature growth stage,

based on which continuous BLG is obtained within 20 min. The BLG is dominated by AB-stacking structure (~61% areal ratio) and exhibits improved mechanical, optical, and electrical properties. Moreover, ~100% AB-BLG single crystal has been successfully grown on the ultraflat single-crystal Cu(111)/sapphire substrate which are prepared by epitaxial deposition of Cu on single-crystal sapphire substrates.’ (Page 11, lines 251-258 in the updated main text)

Meanwhile, all the descriptions about the Cu substrates and BLG samples have been carefully revised to avoid possible misunderstandings, which are listed below.

‘Fig. 1a-c shows the representative optical microscope (OM) images of the as-prepared graphene films that were grown on polycrystalline Cu foils and then transferred onto SiO₂/Si substrates after varied growth time.’ (Page 5, lines 117-120 in the new main text)

‘Electrical quality of the AB-BLG is probed using the dual-gate Hall bar device, where the AB-BLG crystal is encapsulated by two bulk hBN flakes (Supplementary Figs. 28 and 29)⁵⁹, which function as top and bottom dielectrics. An observable bandgap can be induced in our AB-BLG device by applying an out-of-plane electric displacement field^{13,14}.’ (Page 10, lines 233-236 in the new main text)

‘**Device fabrication and measurement.** For the electrical measurement, the AB-BLG is firstly’ (Page 13, line 306 in the new main text)

‘For the photocurrent measurement, the AB-BLG FET device is fabricated’ (Page 13, lines 312-313 in the new main text)

‘**Figure 1 | Fast synthesis of large-area BLG film with the assistance of CO₂.** a-c OM images of the transferred graphene after 1 min (a), 10 min (b), and 20 min (c) growth on commercial polycrystalline Cu foils.’ (Page 19, lines 475-477 in the main text)

‘**Figure 4 | Synthesis of 100% AB-BLG on Cu(111) substrates.** a Distribution of twist angles based on SAED patterns of the BLG grown on the single-crystal Cu(111) film substrate. Inset: Statistical results of the stacking order (AB stacking or non-AB stacking) (center) and STEM image of the AB-BLG with atomic resolution (top right). b Raman spectra of the AB-BLG

transferred onto a SiO₂/Si substrate *acquired via a line scanning with the step of 50 μm by normalizing the G band intensity.* **c** Raman 2D band of the AB-BLG, which is fitted by four Lorentzian peaks. **d** Two-dimensional plot of the ρ_{xx} as functions of both V_{tg} and V_{bg} of a dual-gate AB-BLG device.’ (Page 20 and 21, lines 509-517 in the new main text)

‘**Supplementary Fig. 29. Electrical measurement of the dual gate AB-BLG Hall bar device.** . **a** Schematic of the Hall bar device of the AB-BLG. **b** OM image of the AB-BLG dual gate Hall bar device. **c** Resistivity of the AB-BLG as a function of V_{tg} , which is measured at room temperature (300 K) with zero V_{bg} . **d** Resistivity of the AB-BLG as a function of the carrier concentration.’ (Page 33, last 3-6 lines in the caption of Supplementary Fig. 29)

‘**c-e** Raman mapping results of the I_G (**c**), I_{2D}/I_G (**d**), and I_D/I_G (**e**), with the MLG in the left and the AB-BLG in the right.’ (Page 34, lines 3-4 in the caption of Supplementary Fig. 31)

3. In Fig 4, what kind of BLG authors used to measure sheet resistance (4b) and dual-gate device (4d)? If author used AB-BLG, why sheet is reduced for BLG in comparison with MLG (b)? Authors claim in the main text that they observed the tunability of the band gap, so BLG for this device should be AB-BLG. But I did not see the resistance varied with both V_{bg} and V_{tg} (Fig. 4d). Could author provide I-V curve of Fig. 4d.

Reply: Thanks for the valuable comments. To measure sheet resistance, the transferred BLG films grown on polycrystalline Cu foils was characterized, which comprises ~ 61% AB-BLG and ~39% tBLG. There are several reasons for the observed reduced sheet resistance of BLG in comparison with MLG. Firstly, according to the equation $\sigma = n * e * \mu$, the electrical conductivity (σ) of graphene is decided by both carrier mobility (μ) and carrier concentration (n). For BLG, even though the carrier mobility is decreased in comparison with MLG, the increased number of conduction channels would contribute to the increased carrier concentration and thus increases the electrical conductivity (*J. Supercond. Nov. Magn.* 2017, 30, 1263). The increased electrical conductance of BLG has been widely investigated and was supported by both theoretical simulation and experimental results (*Energy Environ. Sci.* 2013, 6, 108; *Nano Lett.* 2009, 9, 8, 2973; *Nat. Nanotechnol.* 2010, 5, 574; *Adv. Mater.* 2011, 23, 1514; *Nano Lett.* 2009, 9, 12). Secondly, our BLG is high quality and has low defect density, as indicated by the negligible D band in Raman spectra, which also contributes to the improved electrical conductivity (*Acta Phys.-Chim. Sin* 2019, 35, 1000). Thirdly, a significantly lower

sheet resistance in BLG with high twist angles than that in AB-BLG was reported by Yuji Araki *et al.* (*ACS Nano* 2022, 16, 14075), which indicates that the presence of tBLG also contributes to reducing sheet resistance of our BLG film. Finally, the enhanced mechanical property of BLG and the optimized transfer process can reduce the possibility of crack formation during transfer compared with that of MLG, which in turn benefits the reliable measurement of sheet resistance values in large area.

To make our manuscript more informative, we have added above discussions in the updated version (Page 14, last paragraph in the updated supporting information), together with eight new citations as Refer. 18-25.

For the electrical transport measurement, graphene grown on a Cu (111) foil, which was composed of isolated AB-BLG domains and continuous MLG film was utilized (Fig. R5a). After Raman characterization across the whole domain (Fig. R5b), the BLG domain was picked up by a hBN nanoflake to fabricate the hBN/BLG/hBN sandwiched structure (Fig. R5c). Note that for all the Raman spectra, the intensity ratio of 2D to G bands (I_{2D}/I_G) is between 0.5 to 1 and the full width at half maximum of 2D band (FWHM(2D)) value is between 40 to 60 cm^{-1} , confirming that this BLG domain is AB-stacking structure (*Nat. Commun.* 2019, 10, 2809).

Figure R5. OM and Raman characterization of the AB-BLG domain used to fabricate the dual-gate device. a OM image of the AB-BLG domain transferred on a SiO₂/Si substrate. **b** Raman spectra acquired on four different positions marked in (a). **c** OM image of the hBN/BLG/hBN sandwiched structure.

Our previous measurement was conducted in a glovebox filled with Ar atmosphere at 300 K, on which condition the impact of thermal fluctuation and interface contamination cannot be ignored. In specific, impacts on the electrical transport measurement of the BLG from both the polymer residues atop the graphene, introduced during the wet transfer, and the bubbles formed

during the stacking, cannot be avoided, since they severely impact the interface quality between hBN and graphene, especially considering that the density and size of bubbles in the BLG region are much larger than those in the MLG regions (Fig. R5c). To further reduce the impact from the measurement environment and surface/interface contaminations, we have loaded the same device into a vacuum chamber ($\sim 10^{-7}$ Torr) and measured it again at 290 K. This time the opening of band gap has been observed more clearly (Figs. R6 and 7). In detail, when sweeping the bottom-gate voltage (V_{bg}) from -15 V to 50 V and sweeping the top-gate voltage (V_{tg}) from 4 V to -15 V, the resistance reached maximum at highest displacement field region (the top-left and bottom-right), confirming the tunability of bandgap with perpendicular dipole electric field (Fig. R6d). The transfer curves of the AB-BLG when sweeping the V_{bg} at different values of the fixed V_{tg} is given in Fig. R7, in which the maximum resistance attained at the charge neutrality point increases with increasing V_{tg} in both the positive and negative directions, further confirming the opening of the band gap.

Figure R6. Synthesis of AB-BLG on Cu(111) substrates. **a** Distribution of twist angles based on SAED patterns of the BLG grown on the ultraflat single-crystal Cu(111) substrate that was obtained by epitaxial growth of Cu(111) (i.e. Cu(111) film) on annealed c-plane sapphire. Inset: Statistical results of the stacking order (AB stacking or non-AB stacking) (centre) and STEM image of the AB-BLG with atomic resolution (top right). **b** Raman spectra of the AB-BLG transferred onto a SiO₂/Si substrate acquired via a line scanning with the step of 50 μm by normalizing the G band intensity. **c** Raman 2D band of the AB-BLG, which is fitted by four Lorentzian peaks. **d** Two-dimensional plot of the ρ_{xx} as functions of both V_{tg} and V_{bg} of a dual-gate AB-BLG device. **e** Photocurrent distribution in the two graphene/electrode junctions. Inset: OM image of the graphene FET device. Scalebar: 5 μm . **f**

Photocurrent mapping result of the graphene device, in which the MLG is on the left part and the BLG is on the right part.

Figure R7. Transfer curves of the dual gate AB-BLG Hall bar device with varied V_{tg} and V_{bg} .

Following this, the mapping result has been updated in Fig. 4d (Fig. R6). Raman and OM characterization of the AB-BLG (Fig. R5) and transfer curves of the device with varied V_{tg} and V_{bg} (Fig. R7) have also been added in the updated supporting information as Supplementary Figs. 28 and 30, accompanied by relevant discussions (Page 17, last 4-10 lines and Page 18, 2nd paragraph in the updated supporting information). Meanwhile, relevant discussion in the Method part have also been revised, as below.

‘The carrier mobility was measured at 300 K in a glovebox filled with Ar and the gate-dependent transfer curves were measured at 290 K under vacuum ($\sim 10^{-7}$ torr).’ (Page 14, lines 310-312 in the updated main text)

4. To create the point defect for second layer growth, authors employed CO_2 (in the main body of manuscript) and plasma etching (Supplementary Note 2). It seems like, once point defect was created, second layer could be grown underneath first layer. H_2 also can etch graphene to create point defect, Could author attempt to create it by H_2 and grow second layer. I am wondering the role of CO_2 , may be it is not only for etching to create defect.

Reply: Thanks for the valuable comments. We found that H_2 cannot replace CO_2 for fast synthesis of BLG, presumably owing to the much stronger etching ability of CO_2 (*Adv. Mater. Interfaces* 2017, 4, 1601065). The reaction between CO_2 and the atomic carbon in graphene lattice would be rapid, which would create sufficient defect sites in the top graphene layer for the fast diffusion of carbon species to grow the second graphene layer. It has been reported that

H₂ tends to etch graphene from its defective regions, such as the edges (*J. Am. Chem. Soc.* 2010, 132, 14751) or the nucleation sites (*ACS Nano* 2012, 6, 1, 126–132; *ACS Nano* 2014, 8, 12, 12806–12813), while CO₂ attacks both the graphene edge and defect-free plane (*Adv. Mater. Interfaces* 2017, 4, 1601065; *Angew. Chem. Int. Edit.* 2019, 58, 14446). A recent work by Wenqian Yao *et al.* (*Adv. Mater.* 2022, 34, 2108608) also reported that H₂ prefers to etch the second graphene layer rather than the top graphene layer because of the presence of abundant active carbon species atop the first layer and the relatively weaker etching ability of H₂.

To fully address the reviewer's comment, we have conducted additional first-principle simulations to compare the graphene etching ability of CO₂ and H₂ from two aspects, the strength of their interactions with graphene (Fig. R8) and the thermodynamic changes in energy during graphene etching reactions (Fig. R9). First, from the perspective of molecular interactions, the CO₂ on graphene shows a much lower adsorption energy ($E_{\text{ad}} = 0.07$ eV) than that of H₂ ($E_{\text{ad}} = 0.76$ eV) (Fig. R8a,c), indicating that the synthetic graphene surface could preferentially capture CO₂ molecules rather than H₂. Further charge density difference (Fig. R8b,d) and the Crystal Orbital Hamilton Populations (COHP) calculations (Fig. R8e-h) also demonstrate a much stronger chemical interaction between CO₂ and graphene, due to the more charge transfer and lower integrated COHP ($-\text{ICOHP} < -6.0$ eV) values between the adsorbed CO₂ and graphene underneath. Therefore, in comparison with H₂, the CO₂ molecules are more likely to be captured by graphene surface and thus more likely to induce structural deformation and etching reactions driven by electron redistribution. Secondly, from the thermodynamic aspect of etching reaction, the formation energy (E_f) of CO₂ etching graphene is much lower than that of H₂ (Fig. R9), which means that even though H₂ can etch the top graphene layer, it is less energy-favourable than CO₂ and thus is more difficult to happen.

Figure R8. Comparison of the adsorption and interactions of CO₂ and H₂ molecules on graphene. **a** Top view of the CO₂ molecule adsorbed on top of the perfect MLG/Cu. **b** Side view of the charge density difference maps between the CO₂ molecule and the MLG/Cu. **c** Top view of the H₂ molecule adsorbed on top of the perfect MLG/Cu. **d** Side view of the charge density difference maps between the CO₂ molecule and the MLG/Cu. In the charge density difference maps, the blue (yellow) parts represent the deficiency (aggregation) of electrons. **e,f** COHP between the C (**e**) or O (**f**) atoms of the CO₂ molecule and the C atoms in the graphene lattice. **g,h** COHP between the two H atoms of the H₂ molecule and the C atoms in the graphene lattice. The adsorption energy (E_{ad}) of CO₂ and H₂ on graphene surface can be calculated as $E_{ad} = E_{tot} - E_{sub} - E_{mol}$, where the E_{tot} and E_{sub} are the total energies of the MLG/Cu surface with and without adsorbed CO₂ or H₂ molecule, and E_{mol} is the energy of a CO₂ or H₂ molecule, respectively.

Figure R9. Formation energy of different etching products obtained via the graphene etching reactions by CO₂ or H₂.

Moreover, the introduction of trace CO₂ also contributes to the rapid formation of abundant active carbon species by releasing O atoms (*Adv. Mater. Interfaces* 2017, 4, 1601065; *ACS Cent. Sci.* 2022, 8, 394), which can decrease the decomposition energy barriers of carbon source (CH₄) in the high-temperature CVD system (*Nat. Nanotechnol.* 2016, 11, 930). This also explains the phenomenon that the growth rate of the second graphene layer using our CO₂-assisted strategy is higher than that using the mild oxygen plasma treatment strategy demonstrated in our supporting information (Supplementary Fig. 10 in the previously submitted version and Supplementary Fig. 16 in the new version).

To make this work more informative, Figs. R8 and 9 have been added as Supplementary Figs. 4 and 5, accompanied by above discussions in the updated version (Page 4, last 9 lines and Page 5, lines 1-21 in the updated supporting information) with eight new citations as Refers. 1-8.

5. In previous work (*Nano Lett.* 2010, 10, 11, 4702–4707), BLG also could be grown in wafer-scale within few tens of min on Cu substrate. Could authors clarify the advantage of their work compared to this previous work?

Reply: Thanks for the valuable comment. This previous work reported the homogeneous synthesis of BLG on Cu foil (~ 2 in. x 2 in. area) by using only CH₄ with large flux (70 sccm) at high temperature growth stage (1,000 °C, 15 min) and then controlling the cooling rate of 18 °C/min (*Nano Lett.* 2010, 10, 11, 4702). However, little information was provided about the growth mechanism and the domain size of the BLG. In comparison with this work, the advantages of our work are listed below: Firstly, we developed a new synthesis strategy to grow BLG by introducing trace CO₂ during the high-temperature growth stage, which can break the self-limitation growth of graphene on Cu substrates and enable the fast synthesis of BLG films in large area. Secondly, the size of continuous BLG films can reach meter-sized scale in our work, and mass production can be possibly achieved *via* both static batch-to-batch and dynamic roll-to-roll approaches. Thirdly, different from this previous work in which the cooling speed is important, the bilayer coverage on Cu has no dependence on the cooling rate in our work, which could further shorten the cooling time and save the production cost. Fourthly, our BLG exhibits improved quality, as proved by the negligible D band in Raman spectra, while the intensity ratio of D to G bands in this previous work is larger than 0.11 for

BLG. Fifthly, the domain size of our BLG is more controllable and can reach over 50 μm on polycrystalline Cu foils based on optimized parameters, while no information about domain size was included in the previous work. Since high flow rate of CH_4 (70 - 140 sccm) is utilized in previous work, it is highly possible that the BLG domain size is much smaller than that in our work. Last but not least, 100% single-crystal AB-BLG was achieved on Cu(111) substrate in our work. Based on these, we strongly believe that our work has sufficient novelty and obvious advantages compared with this previous work.

6. Is tri-layer or tetra-layer graphene possible using this CO_2 assisted growth? By simple thinking, after finishing 2nd layer growth, CO_2 could be introduced again to etch graphene, leaving space for carbon species diffusing into Cu for tri-layer or tetra-layer graphene growth.

Reply: Thanks for your valuable comment. It is possible to grow tri-layer graphene (TLG) or tetra-layer graphene on Cu with the aid of CO_2 , however, according to our experimental results, the nucleation density, growth rate and coverage of TLG and tetralayer graphene are much lower than those of BLG (Fig. R10a), which makes the large-area synthesis of BLG film possible on Cu after optimizing CVD parameters (Fig. R10b-d).

Figure R10. Graphene films transferred on SiO_2/Si substrates after growth with the aid of CO_2 . **a** OM image of the graphene film composed of MLG, BLG, and multilayer graphene. **b-d** OM images of the BLG in large area acquired after optimizing CVD parameters.

To grow TLG or tetra-layer graphene, abundant defective structures in the bottom graphene layer of the pre-formed BLG or TLG are required so that carbon species could diffuse across the defects and enter the interface between the bottom graphene layer and Cu. However, with the increased graphene thickness, it is more difficult for CO₂ to etch bilayer or multilayer graphene, which is supported by both of our theoretical calculation results.

First principle simulations have been conducted by constructing the MLG, BLG and TLG structures on Cu(111) surface to quantitatively investigate the difference of CO₂ etching ability on multilayer graphene. We know that the etching rate of graphene (V_g) at a specific temperature depends on the etching difficulty of single carbon atoms in graphene (V_c) and the flux of etching gas (F_{CO_2}), scilicet as: $V_g \propto V_c * F_{CO_2}$. Since the chemical environment and coordination structure of all C atoms in the perfect multilayer graphene are the same, we believe that the etching rate of C atoms in each graphene layer is the same and can be set as constant under specific CVD conditions. Therefore, the etching difficulty of graphene with different layers only depends on the effective F_{CO_2} contacting it. Further on, the F_{CO_2} can be calculated by the concentration of CO₂ in contact with graphene of different layers ($\rho = e^{-\frac{E_{ad}}{k_B * T}}$) and the corresponding effective exposure area (S), which can be written as:

$$F_{CO_2} = S * e^{-\frac{E_{ad}}{k_B * T}}$$

where k_B and T are Boltzmann constant and experimental temperature, respectively. As illustrated in Fig. R11a-d, the adsorption energy (E_{ad}) of CO₂ molecules on the MLG, BLG, and TLG surfaces were calculated to be -0.07 eV, -1.44 eV, and -2.27 eV, respectively, which indicates the reduced concentration for CO₂ to be captured by BLG or TLG in comparison with the MLG. At the same time, the exposure area (S) of the bottom graphene layer to CO₂ also decreases with the layer number. Bringing into the numerical calculation, we can obtain the F_{CO_2} values (Fig. R11e). It is clearly observed that the etching flux of graphene decreases significantly with its layer number. Moreover, since the real defect density of BLG and TLG is much lower than that estimated from the calculation model we built here (Fig. R11b,c), the formation of TLG and tetra-layer graphene with the aid of CO₂ is thus even much more difficult.

Figure R11. Theoretical estimation of CO₂ flux on MLG, BLG and TLG. **a** Configurations of CO₂ molecules adsorbed on the MLG (**a**), BLG (**b**) and TLG (**c**) surfaces. **d** Adsorption energy of CO₂ molecules on the MLG (green), BLG (red), and TLG (purple). **e** Comparison of the etching flux of CO₂ on MLG (green), BLG (red), and TLG (purple) surface as a function of temperature.

Our finding is in good agreement with the calculation results by Wu *et al.* (*J. Phys. Chem. C* 2014, 118, 6201). They also reported that the energy barrier for C atoms to penetrate BLG (2.94 eV) is much higher than that to penetrate MLG (1.48 eV), indicating that growing a third layer graphene is much more difficult than growing BLG.

To fully address the reviewer's comments, Fig. R11 has been added in the updated version as Supplementary Fig. 7, together with above discussions (Page 6 and Page 7, lines 1-11 in the updated supporting information).

7. To prove second layer grow underneath first layer, author performed the depth profile mapping (Fig. 2c). After 11-13s sputtering, there is ring-like pattern in ¹²C²⁻. Why this ring-like pattern did not appear in ¹³C²⁻.

Reply: Thanks for your valuable comment. There are mainly three reasons accounting for the unclear ring-like patterns in the ¹³C²⁻ mapping image after 11-13 s sputtering using the time-of-flight secondary ion mass spectrometry (ToF-SIMS). Firstly, it takes more than seven seconds to completely etch each graphene layer by the Bi³⁺ sputtering in our ToF-SIMS experiments (Fig. R12a) while the mapping results shown in Fig. 2g were acquired based on the signals collected only in two seconds (i.e., 11-13 s) (Fig. R12b). As a result, the total signal amount is limited and thus the mapping images are much noisier than the mapping images acquired based on 0-15 s accumulations (Fig. R12c). Secondly, since the m/z of ¹³C²⁻ is equal to that of ¹²CH₂²⁻, the signals of ¹²CH₂²⁻ was also counted into the mapping results of ¹³C²⁻,

while for $^{12}\text{C}^{2-}$, there were not this kind of signal disturbance. Thirdly, the spatial resolution of the ToF-SIMS measurements is equal to that of optical microscope, while the width of each hexagonal-shaped ring in the second graphene layer is less than $2\ \mu\text{m}$ (Fig. R12b-e), which makes it more challenging to clearly see the hexagonal shapes.

To show clearer ring-like patterns of both $^{12}\text{C}^{2-}$ and $^{13}\text{C}^{2-}$, we have added the mapping results of $^{13}\text{C}^{2-}$ and $^{12}\text{C}^{2-}$ based on 0-15 s accumulation (Fig. R12c). Besides, the ring-like patterns of $^{13}\text{C}^{4-}$ and $^{12}\text{C}^{4-}$ acquired based on sputtering time of 11-13 s and 0-15 s also confirm the alternative distribution of ^{13}C and ^{12}C elements in the as-formed BLG domains (Fig. R12d,e), which also agree well with the Raman mapping results (Fig. 2e, f and Supplementary Figs. 7, 8 and 10 in the updated version).

To make this manuscript more informative, Fig. R12 has been added in the updated supporting information as Supplementary Fig. 12, accompanied by relevant discussions (Page 9, lines 5-10 in the updated supporting information)

Figure R12. ToF-SIMS measurement results. **a** Depth profiles for $^{12}\text{C}^{2-}$ (blue) and $^{13}\text{C}^{2-}$ (red) of the BLG with $\sim 50\%$ bilayer coverage. **b,c**, Spatial distribution of $^{12}\text{C}^{2-}$ (top) and $^{13}\text{C}^{2-}$ (bottom) in the scanned regions during the sputter time of 11-13 s (**b**) and 0-15 s (**c**). **d,e** Spatial distribution of $^{12}\text{C}^{4-}$ (top) and $^{13}\text{C}^{4-}$ (bottom) in the scanned regions during the sputter time of 11-13 s (**d**) and 0-15 s (**e**).

8. And in Supplementary Note 3, authors performed mild oxygen plasma etching to etch top layer graphene without destroying the bottom layer. Could author provide optical microscopy image of BLG before and after mild plasma etching to prove layer by layer etching?

Reply: Thanks for the valuable comment. Following the reviewer's suggestion, the OM image of the BLG domain after the mild oxygen plasma treatment has been added, followed by the

corresponding updating in Supplementary Fig. 13 in the new version (Fig. R13). We did not take OM image of the same position before the mild oxygen plasma treatment, because by performing mild oxygen plasma etching, only point defects were formed in the top graphene layer, which is not detectable in the OM. To further prove this, we have conducted the same oxygen plasma etching treatment to another BLG sample and taken the OM images before and after the etching (Fig. R14).

Figure R13. Confirmation of the BLG stacking order. **a** OM image of the BLG domain after the mild oxygen plasma treatment to create point defects in the top graphene layer. **b,c** Raman map results of $^{13}\text{C-I}_G$ (**b**) and $^{12}\text{C-I}_G$ (**c**). **d** Raman spectra of the $^{12}\text{C-MLG}$ (green) and $^{12}\text{C}/^{13}\text{C-BLG}$ (red) after plasma treatment. **e** Raman D bands appeared in $^{12}\text{C-MLG}$ (green) and $^{12}\text{C}/^{13}\text{C-BLG}$ (red) regions after plasma treatment.

Figure R14. BLG before and after the mild oxygen plasma treatment. a,b OM images of the BLG domains before (a) and after (b) treatment.

Minor

1. Fig. 1g, author should point out reference number directly in the panel. It's very difficult to follow up if all reported works was marked as blue circle.

Reply: Thanks for the valuable suggestion to strengthen our manuscript. To help readers follow up, the numbers of cited papers in Fig. 1g have been added according to their reference number (Fig. R15).

Figure R15. Fast synthesis of large-area BLG film with the assistance of CO₂. a-c OM images of the transferred graphene after 1 min (a), 10 min (b), and 20 min (c) growth on commercial

polycrystalline Cu foils. d Relationship between the graphene bilayer coverage and the flow rate of CO₂. Inset: Typical OM images of the graphene grown using 5 sccm (top left) and 20 sccm (bottom right) CO₂. **e** Relationship between the graphene growth time and its bilayer coverage when 30 sccm CO₂ was introduced to grow graphene for 20 min. **f** Relationship between the graphene growth time and its bilayer coverage when 30 sccm CO₂ was utilized only in the first 10 min. Inset: OM image of the synthesized graphene after flowing H₂ and CH₄ for 90 min but only flowing CO₂ in the first 10 min. **g** Relationship between the graphene growth time and its bilayer coverage, showing the advantage of our CO₂-assisted strategy (red) in comparison with previous works (blue). **h** Photographs of the large-area BLG films grown on eight pieces of commercial Cu foils in one batch. **i** Statistic of the graphene bilayer coverage of the graphene samples in (**h**) (red) and the graphene sample grown without CO₂ (green).

2. Line 109: Supplementary Fig. 4 was wrongly mentioned

Reply: Thanks for the kind reminding. We have corrected the citation of ‘*Supplementary Fig. 4*’ to ‘*Supplementary Fig. 1*’ in Line 109 in the main text and carefully checked citations of all other figures in both the main text and supporting information to avoid such mistakes.

3. Too much data in the main Figure. Please make it concisely by moving not important data to Supporting information.

Reply: Thanks to Reviewer #1 for the beneficial comments. Following the reviewers’ constructive suggestions, we have updated the Figs. 3 and 4 (Figs. R16 and 17) in the main text to make it more concise and moved the Raman mapping results of AB-BLG from Fig. 3 in the previous version to Supplementary Fig. 26b (Fig. R18) in the updated supporting information.

Figure R16. Characterization of the continuous BLG grown on polycrystalline Cu foils. **a** HRTEM image of the AB-BLG with lattice resolution. Inset: SAED pattern of the AB-BLG and the intensity profile along the red line. **b** Distribution of twist angles based on SAED patterns of the BLG grown on the Cu(100)-dominated polycrystalline Cu. Inset: Statistical results of the stacking order (AB stacking or non-AB stacking) (centre) and low-magnification SEM image of the BLG transferred onto a TEM grid (top right). **c** AFM nano-indentation measurement result of the BLG. Inset: Fitting of the Young modulus. **d** UV-vis spectra of the MLG (green) and BLG (red) transferred onto quartz substrates. Inset: Large-area transmittance mappings of the MLG (left) and BLG (right) at 550 nm wavelength. **e** Statistical histograms of the sheet resistance of the MLG (green) and BLG (red) transferred onto SiO₂/Si substrates. Inset: photograph of the large-area BLG transferred onto a 4 inch-sized SiO₂/Si substrate. **f** Sheet resistance mapping result of the BLG.

Figure R17. Synthesis of AB-BLG on Cu(111) substrates. **a** Distribution of twist angles based on SAED patterns of the BLG grown on the ultraflat single-crystal Cu(111) substrate that was obtained by epitaxial growth of Cu(111) (i.e. Cu(111) film) on annealed c-plane sapphire. Inset: Statistical results of the stacking order (AB stacking or non-AB stacking) (centre) and STEM image of the AB-BLG with atomic resolution (top right). **b** Raman spectra of the AB-BLG transferred onto a SiO₂/Si substrate acquired via a line scanning with the step of 50 μm by normalizing the G band intensity. **c** Raman 2D band of the AB-BLG, which is fitted by four Lorentzian peaks. **d** Two-dimensional plot of the ρ_{xx} as functions of both V_{tg} and V_{bg} of a dual-gate AB-BLG device. **e** Photocurrent distribution in the two graphene/electrode junctions. Inset: OM image of the graphene FET device. Scalebar: 5 μm. **f** Photocurrent mapping result of the graphene device, in which the MLG is on the left part and the BLG is on the right part.

Figure R18. Raman characterization of the AB-BLG. **a** OM image of the AB-BLG transferred on SiO₂/Si. **b** Statistical results of the FWHM(2D) of AB-BLG. Inset: Mapping of FWHM(2D). **c** Statistical histogram of I_{2D}/I_G. Inset: Mapping result of I_{2D}/I_G.

The relevant discussions have also been revised accordingly, as below.

‘Characterization of the continuous BLG grown on polycrystalline Cu foils.’ (Page 8, line 185 in the new main text)

‘To further characterize the crystallinity of the BLG, large-area HRTEM images of AB-BLG and tBLG with lattice resolution were also captured (Supplementary Fig. 18), and the absence of defects clearly confirms the high quality of the BLG.’ (Page 9, lines 199-201 in the new main text)

‘Synthesis of AB-BLG on Cu(111) substrates.

Based on our CO₂-assisted strategy, two kinds of Cu(111) substrates were prepared as

substrates for growing single-crystal AB-BLG. The ratio of AB-BLG increased to ~96% when using a single-crystal Cu(111) foil substrate which derived from the high-temperature annealing of commercial polycrystalline Cu foils⁵⁷ (Supplementary Fig. 21 and Fig. 22). Moreover, 100% AB-stacking structure has been successfully synthesized on the ultraflat single-crystal Cu(111) that was obtained by epitaxial growth of Cu(111) (i.e. Cu(111) film) on annealed c-plane sapphire⁵⁸ (Fig. 4a and Supplementary Figs. 23-25),’ (Page 10, lines 214-221 in the new main text)

‘Supplementary Fig. 26. OM and Raman characterization of the AB-BLG. a OM image of the BLG transferred to a SiO₂/Si substrate. **b** Statistical results of the FWHM(2D) of AB-BLG. Inset: Mapping of FWHM(2D). **c** Statistical histogram of I_{2D}/I_G. Inset: Mapping result of I_{2D}/I_G.’ (Page 32, 3rd paragraph in the updated supporting information)

Reviewer #2 (Remarks to the Author)

Authors report the fast synthesis of large-area bilayer graphene (BLG) on Cu by introducing trace amount of CO₂ during a high-temperature growth process. Bilayer graphene exhibits tunable twist angle and band structure with high mechanical strength, low resistance, and uniform transmittance. These unique properties hold immense promise for next-generation devices. The manuscript presents a great opportunity to reduce CO₂ and methane into a useful BLG growth for fundamental physics studies. However, there are some challenges with growth mechanism of this work, hence I recommend that the manuscript is published in Nature Communication after minor revision. I left some comments to enhance the quality of this manuscript.

Reply: We appreciate very much for the positive comments from the reviewer #2 on the significance of our work and the explicit recommendation of publication. Following the valuable suggestions from the reviewer #2, we have made point-to-point response and revisions.

Comments.

1. First, a fast synthesis of AB-BLG with the assistance of CO₂ reduction has been reported in Gong et al ACS Cent. Sci. 2022, 8, 394–401. They proposed that CO₂ can simultaneously serve as an etching gas and carbon source for graphene growth. Oxygen atoms provided by CO₂ create defects on monolayer graphene (MLG), which break the MLG self-limiting growth and host the nucleation sites for BLG. Could authors distinguish their work from this work? Does carbon in CO₂ play a role in the graphene formation as reported by Gong et al ACS Cent. Sci. 2022, 8, 394–401 and Seekwaew et al Membranes 2022, 12, 796? Also, did authors confirm the growth of BLG using only trace CO₂ supply during their growth process? Authors should clearly discuss the research trend of CO₂ assistance for BLG growth to enhance the novelty of this work.

Reply: Thanks for the constructive comments from the reviewer to strengthen our manuscript. In our work, rather than acting as the carbon source, CO₂ mainly functions as an etchant to produce abundant defects in the MLG regions; therefore, the carbon species could diffuse through the produced defects to reach the Cu substrate, and subsequently fuel the growth of the second graphene layer underneath. The introduction of CO₂ effectively breaks the self-

limitation growth of graphene on Cu substrate, enabling the fast synthesis of large-area BLG films on Cu substrates. Overall, the C atoms in CO₂ do not function as carbon sources for graphene formation in this work, and thus the role of CO₂ in BLG formation is different from those discussed by Gong *et al.* (*ACS Cent. Sci.* 2022, 8, 394) and Seekwaew *et al.* (*Membranes* 2022, 12, 796). We will point out the novelty and advantages of our work in comparison with above two works one by one as follows.

The work reported by Gong *et al.* (*ACS Cent. Sci.* 2022, 8, 394) utilizes CO₂ as the carbon source to grow large BLG domains on Cu substrates. By using an additional catalyst (Ni/Al₂O₃), the relatively inert CO₂ can be activated to produce more reactive carbon sources. At the same time, they reported that the O atoms decomposed from CO₂ can create defects on SLG and host the nucleation sites for BLG, enabling the breaking of the self-limitation growth of graphene on Cu. It is indicated that the second layer grows atop the first layer in both Fig. 1 and Fig. 4i. The maximum bilayer coverage on Cu substrates was only 65%, which was achieved after 40 min growth. In contrast, in our work, the decomposition of inert CO₂ can be avoided for initiating the growth of BLG, and the second graphene layer grows beneath the first graphene layer, as confirmed by the ToF-SIMS measurement (Fig. 2g). Therefore, no additional catalyst was required to activate the CO₂ in our work, and the simple mixing of CO₂, CH₄, and H₂ gases would enable the growth of BLG which demonstrates fine scalability and tunability of the BLG growth process. Moreover, owing to the facile production of defective sites in the top graphene layer enabled by CO₂ and the continuous supply of carbon species via diffusion through these defects, 100% bilayer coverage can be achieved within 20 min growth on Cu substrates. Therefore, our work can achieve the growth of BLG with higher bilayer coverage in a fast manner in comparison with the previous work reported by Gong *et al.* (*ACS Cent. Sci.* 2022, 8, 394).

The work conducted by Seekwaew *et al.* (*Membranes* 2022, 12, 796) requires a H₂ plasma generator loaded upstream the CVD system to pre-treat Cu substrates before growing graphene using CO₂ as the carbon source. By controlling the H₂ plasma power, the graphene thickness can be tuned from two to six layers. However, both the obvious D band in the Raman spectra of obtained multilayer graphene, and the dense distribution of wrinkles in the corresponding SEM images, as well as the unclean graphene surface in TEM images, indicate that the relatively reduced quality of the obtained BLG. In contrast, with our CO₂-assisted method, the Cu foils requires no special pre-treatment; CH₄ rather than CO₂ is employed as the carbon

source; and the BLG has exhibited much higher crystallinity, as confirmed by the noise-level D band intensity.

To address the reviewer's comments, we have conducted more experiments to investigate the role of CO₂ in rapid synthesis of BLG. Following the reviewer's suggestions, we have grown graphene by only introducing trace CO₂ (5 sccm); however, no graphene film was formed on the Cu surface even after 60 min growth (Fig. R19a). We have also introduced both CO₂ (5 sccm) and H₂ (500 sccm) during the graphene growth process but still failed to obtain graphene (Fig. R19b). Instead, the introduction of CO₂ (5 sccm), H₂ (500 sccm) and CH₄ (10 sccm) gases together, could enable the growth of continuous graphene films (Fig. R19c). To further prove that the C atoms from CO₂ was not used for the graphene growth, we used ¹³C-labeled CO₂ (¹³CO₂, 30 sccm) and ¹²C-labeled CH₄ (¹²CH₄, 5 sccm), together with H₂ (500 sccm) to grow graphene. However, even though the flow rate ratio of ¹³CO₂ to ¹²CH₄ is high (6:1), we still only obtained ¹²C-labeled graphene (Fig. R20).

Figure R19. OM images of the Cu foil after high-temperature CVD processes. a-c OM images of the Cu foils after using trace CO₂ (a), mixing gases of CO₂ and H₂ (b) and mixing gases of CO₂, CH₄ and H₂ (c) to grow graphene at 1,020 °C for 60 min.

Figure R20. OM and Raman characterization of the BLG grown using $^{12}\text{CH}_4$, H_2 and $^{13}\text{CO}_2$. **a** OM image of the BLG transferred to a SiO_2/Si substrate. **b** Raman spectra acquired from 4 different positions marked in **(a)**. **c** Mapping result of the G band position of the squared-shaped region marked in **(a)**. **d** Statistical result of the G band position.

In addition, the ring-like patterns of ^{12}C and ^{13}C elements in both the ToF-SIMS and Raman measurement results (Fig. 2e-g) when $^{12}\text{CO}_2$ and $^{13}\text{CH}_4/^{12}\text{CH}_4$ are used for BLG growth also exclude the contributions of C atoms from CO_2 to graphene growth. Therefore, the C atoms in CO_2 cannot play a role in the graphene formation as reported by Gong *et al.* (*ACS Cent. Sci.* 2022, 8, 394) and Seekwaew *et al.* (*Membranes* 2022, 12, 796).

To make our manuscript more informative, relevant discussions about the role of CO_2 have been added in the updated supporting information, as below.

‘Note that when CO_2 or the mixing gases of CO_2 and H_2 flow into the CVD system, no graphene film can be synthesized, indicating that CO_2 cannot function as the carbon source to grow graphene in this work.’ (Page 4, lines 1-3 in the updated supporting information)

‘In addition, the ring-like patterns of ^{12}C and ^{13}C elements in both the ToF-SIMS and Raman measurement results also exclude the contributions of C atoms from CO_2 to graphene growth, even though the flux of CO_2 is much higher than that of CH_4 .’ (Page 10, lines 1-4 in the updated supporting information)

Following the reviewer's constructive suggestions, to enhance the novelty of this work, clear discussions on the research trend of CO₂ assistance for BLG growth have been added in the updated manuscript, together with six new citations as Refers. 21, 26-30, as below.

'CO₂, the greenhouse gas, has been widely utilized to produce valuable carbon-based nanomaterials to mitigate the adverse effects of high CO₂ emissions. Recently, the capability of CO₂ for high-temperature CVD growth of graphene has been reported by several groups^{26,27}. In detail, CO₂ can be utilized for the pre-treatment of commercial Cu substrates based on its selective etching ability of the disordered carbon impurities^{26,27} or for growing graphene as the carbon source²⁸. However, when CO₂ is employed as carbon source, additional catalyst, such as Ni/Al₂O₃^{21,28} and CuPd²⁹, or special treatment of the substrates³⁰ are usually required, which might hinder the compatibility with the mass production processes. In all, even though isolated MLG domains²⁸, continuous MLG films^{26,27}, and isolated BLG domains²¹ have been successfully grown based on CO₂, till now, the controlled preparation of large-area high-quality continuous BLG on Cu has not been achieved yet.' (Page 4, lines 88-97 in the updated main text)

2. Apart from the penetrative mechanism of second layer graphene as suggested by *J. Phys. Chem. C* 2014, 118, 6201–6206, is there the possibility of defects generation and formation of second growth on top of the first layer during growth process?

Reply: Thanks for the valuable comment. According to the previous work published by Huy Q Ta *et al.*, it is possible to form the second layer graphene on top of the first layer when the CH₄ to H₂ ratios is relatively high. This usually results in the irregular shape of the second graphene layer and formation of a larger portion of non-AB BLG. In contrast, with relatively lower CH₄ to H₂ ratios, the second graphene layer was usually formed underneath the top layer, accompanied by the more regular and hexagonal and a high ratio of AB-stacking structure (*Nano Lett.* 2016, 16, 10, 6403). The latter is in good agreement with the results observed in our work, suggesting that our second graphene layer is grown underlying the top graphene layer.

Stacking sequence of the BLG was identified by performing ToF-SIMS and Raman characterizations in our work (*Nano Lett.* 2013, 13, 2, 486). Note that, BLG sample with ~ 50% bilayer coverage was used to easily distinguish the MLG and BLG regions, and the sample was

synthesized via alternately introducing $^{13}\text{CH}_4$ and $^{12}\text{CH}_4$ every 2 min for 10 min. Firstly, mild Bi^{3+} sputtering was conducted to acquire the depth profile of ^{12}C and ^{13}C using ToF-SIMS. As indicated, the top graphene layer is mainly composed of ^{12}C , while the bottom graphene layer is composed of $\sim 50\%$ ^{12}C and $\sim 50\%$ ^{13}C (Fig. R21), verifying that the second-layer graphene grows underneath the first-layer graphene. Meanwhile, Raman characterization of the $^{13}\text{C}/^{12}\text{C}$ -BLG treated after mild etching using oxygen plasma was conducted (Fig. R22). Appearance of ^{12}C -labelled D band in both the MLG and BLG regions and absence of ^{13}C -labelled D band (Fig. R22e) further implied that the second graphene layer is grown beneath the first graphene layer

Figure R21. ToF-SIMS measurement results. **a** Schematic of the CO_2 -assisted BLG growth via the alternative injection of $^{12}\text{CH}_4$ and $^{13}\text{CH}_4$. **b** Depth profiles for $^{12}\text{C}^{2-}$ (blue) and $^{13}\text{C}^{2-}$ (red) of the BLG with $\sim 50\%$ bilayer coverage. **c-e**, Spatial distribution of $^{12}\text{C}^{2-}$ (top) and $^{13}\text{C}^{2-}$ (bottom) in the scanned regions during the sputter time of 0-2 s (**c**), 11-13 s (**d**) and 0-15 s (**e**). **f,g** Spatial distribution of $^{12}\text{C}^{4-}$ (top) and $^{13}\text{C}^{4-}$ (bottom) in the scanned regions during the sputter time of 11-13 s (**f**) and 0-15 s (**g**). Scale bar in (**c-g**): 5 μm .

Figure R22. Confirmation of the BLG stacking order. **a** OM image of the BLG after the mild oxygen plasma treatment. **b,c** In situ Raman map results of ^{13}C -I_G (**b**) and ^{12}C -I_G (**c**). **d** Raman spectra of the ^{12}C -MLG (green) and $^{12}\text{C}/^{13}\text{C}$ -BLG (red) after plasma treatment. **e** Raman D bands appeared in ^{12}C -MLG (green) and $^{12}\text{C}/^{13}\text{C}$ -BLG (red) regions after plasma treatment.

3. On page 8, lines 168-17: authors stated, “Note that, for AB-BLG, the hexagonal SAED pattern with diffraction intensity ratio of the outer {1-210} peak over the inner {1-110} peak is 2:1 (Fig. 3a)”. However, in Fig. 3a, labelling reads (1-210 and 0-110). Authors should clarify this.

Reply: Thanks for the kind reminding. We feel very sorry for making this mistake. The content in the main text has been revised from {1-110} to {0-110} (Page 9, line 194 in the new main text).

Reviewer #3 (Remarks to the Author)

This paper “Fast synthesis of large-area bilayer graphene film on Cu” by J. Zhang et al., reports the controlled synthesis of AB-stacked BLG by introducing CO₂ gas in CH₄ feedstock. The authors claim that they can grow uniform BLG in short time (20 min). In addition, centimeter or meter-scale BLG can be also synthesized by large Cu foils. It seems that CO₂ acts as weak oxidant that makes holes to grow the second graphene layer underneath the monolayer graphene. Their finding is interesting and important.

However, many papers have been already published about the growth of uniform, AB-stacked BLG by CVD. Although the finding of CO₂-assisted BLG is interesting, I feel that the findings are not new enough to be published in Nature Communications. In addition, as listed below, there are many weaknesses in this manuscript. Therefore, I suggest publication in more specialized journals.

Reply: We sincerely thank the reviewer #3 for his/her efforts in reviewing our manuscript and are very grateful for the positive evaluation that ‘*Their finding is interesting and important*’. To address the reviewer’s concern, we have gone back to the lab, and conducted additional experiments and theoretical simulations in the past three months. We would like to point out that even though the growth of uniform, stacked AB-BLG has been reported in recent years, all the work requires the preparation of Cu-based alloy with precise composition, such as CuNi (*Nat. Nanotechnol.* 2020, 15, 289) and CuSi (*Nat. Nanotechnol.* 2020, 15, 861), which would hinder the mass productions, because the industrial production of large-area CuNi alloy foils remain not possible. Furthermore, currently, the growth of large-area BLG films on commercially available Cu foil is still hindered by limited bilayer coverage (*ACS Cent. Sci.* 2022, 8, 394) or poor crystallinity of the continuous graphene film (*Membranes* 2022, 12, 796; *Nano Lett.* 2010, 10, 11, 4702–4707). In clear contrast, in our work, the meter-sized BLG, which exhibited low defect density, enhanced mechanical strength, reduced sheet resistance and uniform optical transmittance, can be prepared within 20 min by using the CO₂-assisted strategy to break the self-limitation growth of graphene on Cu. Moreover, 100% AB-BLG single crystal has been also successfully synthesized on large-area Cu(111) substrates. Therefore, based on this novel BLG growth mechanism and the significant breakthrough in the domain size, scalability and received quality of BLG, we sincerely hope that Reviewer #3 will support the acceptance of our work by *Nature Communications*.

1. There are many papers reporting the CVD growth of bilayer, monolayer, and few-layer graphene using CO₂ gas. However, in this manuscript, the authors cited none of these papers, which is not fair.

- P. Gong et al., *Precise CO₂ Reduction for Bilayer Graphene*, ACS Central Science 8, 394, (2022)

- Y. Seekaew et al., *Conversion of Carbon Dioxide into Chemical Vapor Deposited Graphene with Controllable Number of Layers via Hydrogen Plasma Pre-Treatment*, Membranes, 12, 796 (2022)

- A. J. Strudwick et al., *Chemical Vapor Deposition of High-Quality Graphene Films from Carbon Dioxide Atmospheres*, ACS Nano 2015, 9, 1, 31 (2015).

- A. K. Grebenko et al., *High-Quality Graphene Using Boudouard Reaction*, Adv. Sci. 9, 2200217 (2022).

- M. K. Kairi et al., *Co-synthesis of large-area graphene and syngas via CVD method from greenhouse gases*, Mater. Lett. 227, 132 (2018).

Reply: Thanks for the valuable comments. We have carefully read the mentioned references, and we would like to point out that our work is different from the previous papers regarding the growth mechanism and quality of as-obtained BLG. In details, in some of the mentioned works CVD growth of graphene was achieved by the decomposition of CO₂ gas to fuel the growth as carbon source, while in the other papers the selective etching ability of CO₂ to remove disordered carbon was used to improve the graphene quality, rather than to grow BLG. However, in our work, a trace amount of CO₂ is introduced during the graphene growth stage to etch the pre-formed MLG and thus create the diffusion channels for carbon species for growing BLG. By mixing CO₂ with CH₄ and H₂, large-area BLG films were successfully synthesized on Cu substrates. Moreover, 100% AB-BLG can be prepared using the ultraflat Cu(111) substrate. More detailed discussions about the novelty and advantages of our work in comparison with these previous papers are given below.

The work reported by Gong *et al.* (*ACS Cent. Sci.* 2022, 8, 394) utilizes CO₂ as the carbon source to grow large BLG domains on Cu substrates. By using an additional catalyst (Ni/Al₂O₃), the relatively inert CO₂ can be activated to produce more reactive carbon sources. At the same time, they reported that the O atoms decomposed from CO₂ can create defects on SLG and host the nucleation sites for BLG, enabling the breaking of the self-limitation growth of graphene

on Cu. It is indicated that the second layer grows atop the first layer in both Fig. 1 and Fig. 4i. The maximum bilayer coverage on Cu substrates was only 65%, which was achieved after 40 min growth. In contrast, in our work, the decomposition of inert CO₂ can be avoided for initiating the growth of BLG, and the second graphene layer grows beneath the first graphene layer, as confirmed by the ToF-SIMS measurement (Fig. 2g). Therefore, no additional catalyst was required to activate the CO₂ in our work, and the simple mixing of CO₂, CH₄, and H₂ gases would enable the growth of BLG which demonstrates fine scalability and tunability of the BLG growth process. Moreover, owing to the facile production of defective sites in the top graphene layer enabled by CO₂ and the continuous supply of carbon species via diffusion through these defects, 100% bilayer coverage can be achieved within 20 min growth on Cu substrates. Therefore, our work can achieve the growth of BLG with higher bilayer coverage in a fast manner in comparison with the previous work reported by Gong *et al.* (**ACS Cent. Sci.** 2022, 8, 394).

The work conducted by Seekwaew *et al.* (**Membranes** 2022, 12, 796) requires the H₂ plasma generator loaded upstream the CVD system to pre-treat Cu substrates before growing graphene, and CO₂ was used as the carbon source for growing graphene. By controlling the H₂ plasma power, the graphene thickness can be tuned from two to six layers. However, the obvious D band in the corresponding Raman spectra, and the dense distribution of wrinkles in SEM images, as well as the unclean graphene surface in TEM images, indicate the relatively low quality of as-obtained BLG. In contrast, with our CO₂-assisted method, the Cu foils requires no special pre-treatment, and CH₄ rather than CO₂ was employed as the carbon source for growing the BLG with higher crystallinity, as confirmed by noise-level D band in Raman spectra.

The work reported by A. J. Strudwick *et al.* (**ACS Nano** 2015, 9, 1, 31) first utilizes CO₂ during the high-temperature annealing process to clean Cu substrates and then uses CO₂ together with CH₄ to grow MLG, and CO₂ was repeatedly introduced into the growth chamber every 10 min to remove the as-formed disordered carbon on graphene surface. Note that the authors claim that CO₂ was used to etch the disordered carbon in both annealing and growth stages. As a result, continuous MLG films was synthesized on low-cost commercial Cu foils without using H₂ in the whole CVD process. In our work, CO₂ is not only used to etch the disordered carbon on Cu surfaces, but also introduced during the graphene growth stage to etch the pre-formed continuous MLG; therefore, the fast diffusion of abundant carbon species would fuel the BLG

growth. Instead of pulse injection, the trace CO₂ was continuously introduced during the entire graphene growth stage, together with CH₄ and H₂, which enables the fast synthesis of large-area high-quality BLG films.

The work conducted by A. K. Grebenko *et al.* (*Adv. Sci.* 2022, 9, 2200217) also only focuses on monolayer graphene growth on Cu rather than BLG growth. In their work, CO₂ is firstly utilized to clean Cu surface before graphene growth. However, after that, the mixing gases of CO₂ and CO are used for graphene growth, where CO functions as the carbon source and CO₂ is used to decrease the graphene nucleation density. As a result, they obtained mm-sized MLG crystals on the Cu substrate. In our work, we also used a high-temperature CO₂ annealing process to clean Cu surfaces. However, we used the combination of H₂, CO₂, and CH₄ to grow high-quality BLG, rather than MLG. In our work, the role of CO₂ was to etch the pre-formed continuous MLG to enable the fast diffusion of abundant carbon species for BLG growth. As a result, continuous BLG with negligible D band was successfully synthesized in our work.

The work conducted by M. K. Kairi *et al.* (*Mater. Lett.* 2018, 227, 132) uses Ni foils as the growth substrate and CO₂ as the carbon source to grow few-layer graphene. In contrast, we focused on the BLG on Cu substrates in this work by using CH₄, rather than CO₂, as the carbon source, where CO₂ was employed to etch the pre-formed MLG and thus create diffusion channels for carbon species to grow BLG.

Overall, our work is clearly distinguished from the previous works using CO₂ for graphene growth and thus has high novelty in the fast synthesis of continuous BLG films with fine controllability and scalability.

Following the reviewer's suggestion, to enhance the novelty of this work, discussion on the research trend of CO₂ assistance for graphene growth have been added in the updated manuscript, together with six new citations as Refers. 21, 26-30, as below.

'CO₂, the greenhouse gas, has been widely utilized to produce valuable carbon-based nanomaterials to mitigate the adverse effects of high CO₂ emissions. Recently, the capability of CO₂ for high-temperature CVD growth of graphene has been reported by several groups^{26,27}. In detail, CO₂ can be utilized for the pre-treatment of commercial Cu substrates based on its selective etching ability of the disordered carbon impurities^{26,27} or for growing graphene as

*the carbon source*²⁸. However, when CO₂ is employed as carbon source, additional catalyst, such as Ni/Al₂O₃^{21,28} and CuPd²⁹, or special treatment of the substrates³⁰ are usually required, which might hinder the compatibility with the mass production processes. In all, even though isolated MLG domains²⁸, continuous MLG films^{26,27}, and isolated BLG domains²¹ have been successfully grown based on CO₂, till now, the controlled preparation of large-area high-quality continuous BLG on Cu has not been achieved yet.’ (Page 4, lines 88-97 in the updated main text)

2. The authors’ claim of “100% AB” in the abstract is not solid. They concluded “100% AB” by measuring 100 electron diffraction patterns (Fig. 3c), but it is not clearly written how and where they measured the diffraction patterns. If they measured one TEM specimen (3 mm) or some of holes in one grid, the conclusion is not so reliable. More careful investigation combining other methods, such as Raman, as well as detailed explanation of the method of the AB ratio determination is required. More importantly, in Fig. 1b, not all the hexagonal BLG grains are not aligned. It is not clear whether the authors used Cu(100) foil or Cu(111) foil in Fig. 1b, but this is strange for me. If Fig. 1b is transferred from a Cu(100) foil, the authors should present optical images of aligned BLG grains transferred from a Cu(111) foil or a Cu(111) film as well.

Reply: We appreciate very much for the valuable comments from Reviewer 3# to strengthen our manuscript. The OM image shown in Fig. 1b was acquired using a BLG sample grown on the Cu(100)-dominated polycrystalline Cu foil. In this work, Figs. 1 and 2 focus on the fast synthesis of large-area BLG films on widely used commercial polycrystalline Cu foils and the growth mechanism investigations. To clear the confusion, we have added more information about the BLG samples discussed in different Figures, as below.

‘Fig. 1a-c shows the representative optical microscope (OM) images of the as-prepared graphene films that were *grown on polycrystalline Cu foils and then* transferred onto SiO₂/Si substrates after varied growth time.’ (Page 5, lines 117-120 in the new main text)

‘Synthesis of AB-BLG on Cu(111) substrates.

Based on our CO₂-assisted strategy, two kinds of Cu(111) substrates were prepared as substrates for growing single-crystal AB-BLG. The ratio of AB-BLG increased to ~96% when using a single-crystal Cu(111) foil substrate which derived from the high-temperature

annealing of commercial polycrystalline Cu foils⁵⁷ (Supplementary Fig. 21 and Fig. 22). Moreover, 100% AB-stacking structure has been successfully synthesized on the ultraflat single-crystal Cu(111) that was obtained by epitaxial growth of Cu(111) (i.e. Cu(111) film) on annealed c-plane sapphire⁵⁸ (Fig. 4a and Supplementary Figs. 23-25),’ (Page 10, lines 214-221 in the new main text)

‘Electrical quality of the AB-BLG is probed using the dual-gate Hall bar device, where the AB-BLG crystal is encapsulated by two bulk hBN flakes (Supplementary Figs. 28 and 29)⁵⁹, which function as top and bottom dielectrics. An observable bandgap can be induced in our AB-BLG device by applying an out-of-plane electric displacement field^{13,14}.’ (Page 10, lines 233-236 in the new main text)

‘Device fabrication and measurement. For the electrical measurement, the AB-BLG is firstly’ (Page 13, line 306 in the new main text)

‘For the photocurrent measurement, the AB-BLG FET device is fabricated’ (Page 13, lines 312-313 in the new main text)

‘Figure 1 | Fast synthesis of large-area BLG film with the assistance of CO₂. a-c OM images of the transferred graphene after 1 min (a), 10 min (b), and 20 min (c) growth on commercial polycrystalline Cu foils.’ (Page 19, lines 475-477 in the main text)

‘Figure 4 | Synthesis of 100% AB-BLG on Cu(111) substrates. a Distribution of twist angles based on SAED patterns of the BLG grown on the single-crystal Cu(111) film substrate. Inset: Statistical results of the stacking order (AB stacking or non-AB stacking) (center) and STEM image of the AB-BLG with atomic resolution (top right). b Raman spectra of the AB-BLG transferred onto a SiO₂/Si substrate acquired via a line scanning with the step of 50 μm by normalizing the G band intensity. c Raman 2D band of the AB-BLG, which is fitted by four Lorentzian peaks. d Two-dimensional plot of the ρ_{xx} as functions of both V_{tg} and V_{bg} of a dual-gate AB-BLG device.’ (Page 20 and 21, lines 509-517 in the new main text)

‘Supplementary Fig. 29. Electrical measurement of the dual gate AB-BLG Hall bar device. . a Schematic of the Hall bar device of the AB-BLG. b OM image of the AB-BLG dual gate Hall bar device. c Resistivity of the AB-BLG as a function of V_{tg}, which is measured at room

temperature (300 K) with zero V_{bg} . **d** Resistivity of the **AB-BLG** as a function of the carrier concentration.’ (Page 33, last 3-6 lines in the caption of Supplementary Fig. 29)

‘**c-e** Raman mapping results of the I_G (**c**), I_{2D}/I_G (**d**), and I_D/I_G (**e**), with the MLG in the left and the **AB-BLG** in the right.’ (Page 34, lines 3-4 in the caption of Supplementary Fig. 31)

Following the reviewer’s suggestions, we went back to the lab and captured OM images of the isolated BLG domains grown on Cu(111) substrates. In specific, the isolated BLG domains grown on the Cu(111) foils exhibit near-hexagonal shapes (Fig. R23) and good alignment while the BLG domains grown on the **ultraflat single-crystal Cu(111) substrate that was obtained by epitaxial growth of Cu(111) (i.e. Cu(111) film) on annealed c-plane sapphire** do not have very regular shape (Fig. R24), which might be owing to the atmosphere-pressure growth (*Small* 2019, 15, 1805395).

Figure R23. OM image of the isolated BLG domains grown on a Cu(111) foil.

Figure R24. OM image of the isolated BLG domains grown on a Cu(111) film prepared on c-sapphire.

To make our work more informative, Figs. R23 and 24 have been added as Supplementary Figs. 23 and 25 in the updated supporting information, together with relevant descriptions, as below.

‘OM image of isolated BLG domains grown on Cu(111) foil was also acquired after transferring graphene onto SiO₂/Si substrate, considering rough surface of the Cu(111) foils. After that, well aligned isolated BLG domains with near-hexagonal shapes can be clearly observed (Supplementary Fig. 22).’ (Page 16, lines 4-7 in the updated supporting information)

‘In addition, the OM image of the isolated BLG domains grown on the Cu(111)/sapphire substrate was captured directly without graphene transfer (Supplementary Fig. 25). Note that the irregular shape of graphene domains might be owing to the atmosphere growth pressure.’ (Page 16, last 6-9 lines in the updated supporting information)

Furthermore, to avoid misunderstanding, we have reorganized Figs. 3 and 4 in the main text to separate the experimental results of the BLG films according to the type of Cu substrates. Therefore, in the updated version, following Figs. 1 and 2, Fig. 3 only discusses the structure and corresponding characterization of the large-area BLG film synthesized on commercial polycrystalline Cu foils (Fig. R25) while Fig. 4 focuses on the preparation and relevant characterizations of the AB-BLG grown on Cu(111) substrates (Fig. R26).

Figure R25. Characterization of the continuous BLG grown on polycrystalline Cu foils. **a** HRTEM image of the AB-BLG with lattice resolution. Inset: SAED pattern of the AB-BLG and the intensity profile along the red line. **b** Distribution of twist angles based on SAED patterns of the BLG grown on the Cu(100)-dominated polycrystalline Cu. Inset: Statistical results of the stacking order (AB stacking or non-AB stacking) (centre) and low-magnification SEM image of the BLG transferred onto a TEM grid (top right). **c** AFM nano-indentation measurement result of the BLG. Inset: Fitting of the Young

modulus. **d** UV-vis spectra of the MLG (green) and BLG (red) transferred onto quartz substrates. Inset: Large-area transmittance mappings of the MLG (left) and BLG (right) at 550 nm wavelength. **e** Statistical histograms of the sheet resistance of the MLG (green) and BLG (red) transferred onto SiO₂/Si substrates. Inset: photograph of the large-area BLG transferred onto a 4 inch-sized SiO₂/Si substrate. **f** Sheet resistance mapping result of the BLG.

Figure R26. Synthesis of AB-BLG on Cu(111) substrates. **a** Distribution of twist angles based on SAED patterns of the BLG grown on the ultraflat single-crystal Cu(111) substrate that was obtained by epitaxial growth of Cu(111) (i.e. Cu(111) film) on annealed c-plane sapphire. Inset: Statistical results of the stacking order (AB stacking or non-AB stacking) (centre) and STEM image of the AB-BLG with atomic resolution (top right). **b** Raman spectra of the AB-BLG transferred onto a SiO₂/Si substrate acquired via a line scanning with the step of 50 μm by normalizing the G band intensity. **c** Raman 2D band of the AB-BLG, which is fitted by four Lorentzian peaks. **d** Two-dimensional plot of the ρ_{xx} as functions of both V_{ig} and V_{bg} of a dual-gate AB-BLG device. **e** Photocurrent distribution in the two graphene/electrode junctions. Inset: OM image of the graphene FET device. Scalebar: 5 μm. **f** Photocurrent mapping result of the graphene device, in which the MLG is on the left part and the BLG is on the right part.

The relevant discussions have also been updated accordingly, as below.

‘Characterization of the continuous BLG grown on polycrystalline Cu foils.’ (Page 8, line 185 in the new main text)

‘To further characterize the crystallinity of the BLG, large-area HRTEM images of AB-BLG and tBLG with lattice resolution were also captured (Supplementary Fig. 18), and the absence

of defects clearly confirms the high quality of the BLG.' (Page 9, lines 199-201 in the new main text)

'Synthesis of AB-BLG on Cu(111) substrates.

Based on our CO₂-assisted strategy, two kinds of Cu(111) substrates were prepared as substrates for growing single-crystal AB-BLG. The ratio of AB-BLG increased to ~96% when using a single-crystal Cu(111) foil substrate which derived from the high-temperature annealing of commercial polycrystalline Cu foils⁵⁷ (Supplementary Fig. 21 and Fig. 22). Moreover, 100% AB-stacking structure has been successfully synthesized on the ultraflat single-crystal Cu(111) that was obtained by epitaxial growth of Cu(111) (i.e. Cu(111) film) on annealed c-plane sapphire⁵⁸ (Fig. 4a and Supplementary Figs. 23-25),' (Page 10, lines 214-221 in the new main text)

In this work, we mainly used transmission electron microscopy (TEM) to evaluate the ratio of AB-BLG in our BLG samples grown on Cu foils. To make a fair evaluation, more than 50 selected area electron diffraction (SAED) patterns were acquired in different positions across the whole 3 mm-sized TEM grid, and the characterized areas were distributed uniformly in the 3 mm region. Based on obtained orientations of the SAED patterns, the ratio of AB-BLG and twisted BLG (tBLG) can be calculated accordingly. Note that, for AB-BLG, the hexagonal SAED pattern with diffraction intensity ratio of the outer peak {1-210} over the inner peak {0-110} is 2:1, while for tBLG, there are two groups of MLG SAED patterns, whose intersection angle corresponds to the twist angle of the tBLG. The statistical result of 100% AB-BLG grown on ultraflat single-crystal Cu(111) substrate that was obtained by epitaxial growth of Cu(111) (i.e. Cu(111) film) on annealed c-plane sapphire was then shown in Fig. 3c in the originally submitted version.

To further confirm the 100% AB-stacking structure of our BLG grown on ultraflat single-crystal Cu(111) substrate on annealed c-plane sapphire, we have conducted LEED in cm-sized regions. The single set of hexagonal diffraction spots with no rotational misalignment suggests the epitaxial growth of AB-BLG on the single-crystal Cu(111) substrate. The identical orientations of the LEED patterns acquired at three different positions across the whole sample (2 x 2 cm²) further demonstrate the successful synthesis of the large-scale AB-BLG single-crystal film in at least cm-sized regions. This provides the larger-region (at least cm-sized regions) characterization of the stacking order of BLG than TEM (Fig. R27).

Figure R27. LEED patterns of the AB-BLG film grown on a Cu(111)/sapphire substrate. a Photograph of the cm-sized BLG/Cu(111)/sapphire sample. **b-d** LEED patterns acquired on the AB-BLG/Cu(111) sample.

Following the suggestions from reviewer 3#, Raman characterization has also been conducted to evaluate the AB-stacking structures after transferring the BLG samples onto SiO₂/Si substrates. Note that the Raman spectra were acquired every 1 mm on three cm-sized BLG samples grown on Cu(111)/Sapphire substrates. For all the spectra, the I_{2D}/I_G value is between 0.5 to 1 and the FWHM(2D) value is between 40 to 60 cm⁻¹, furthering confirming the successful preparation of AB-BLG (100% ratio) over the large area (Fig. R28).

Figure R28. Raman characterization of the AB-BLG samples. a-c Raman spectra of sample 1 (b),

sample 2 (b), and sample 3 (c) acquired every 1 mm.

To further strengthen our manuscript, Figs. R27 and 28 have been added as Supplementary Figs. 24 and 27 in the new version. Detailed explanations of the method used to determine AB-ratio and relevant discussion about these new experimental results have also been added in the updated supporting information, as following.

‘For each sample, more than 50 selected area electron diffraction (SAED) patterns were acquired on different positions across the whole region of the 3 mm-sized TEM grid across the whole 3 mm-sized TEM grid, which distribute uniformly in large area, rather than concentrating on some special regions.’ (Page 12, lines 7-10 in the updated supporting information)

‘TEM and SAED characterizations were firstly performed to evaluate the stacking structures of the BLG film grown on Cu(111) substrates. Note that SAED patterns were acquired across the whole TEM grid to evaluate the mono-crystallinity of the synthesized BLG film and the ratio of AB-stacking BLG (AB-BLG) structure increased from <65% on polycrystalline Cu to ~96% (Supplementary Fig. 21) by using the Cu(111) foils prepared via high-temperature annealing²⁶.’ (Page 15, last 2 lines and Page 16, lines 1-3 in the updated supporting information)

‘The continuous BLG film grown on the ultraflat single-crystal Cu(111) substrate that was obtained by epitaxial growth of Cu(111) (i.e. Cu(111) film) on annealed c-plane was also transferred to TEM grid for SAED characterizations. 100% AB-BLG was obtained on this substrate (Supplementary Fig. 23).’ (Page 16, lines 8-11 in the updated supporting information)

‘Moreover, low energy electron diffraction (LEED) patterns of the AB-BLG film grown on the Cu(111)/sapphire substrate were also acquired (Supplementary Fig. 24). The single set of hexagonal diffraction spots with no rotational misalignment suggests the epitaxial growth of AB-stacking BLG (AB-BLG) on the single-crystal Cu(111) substrate. The identical orientations of the LEED patterns acquired at three different positions across the whole sample (2 x 2 cm²) further demonstrate the successful synthesis of the large-scale AB-BLG single-crystal film in at least cm-sized regions.’ (Page 16, lines 11-17 in the updated supporting information)

‘To further evaluate the AB-stacking structures of the BLG grown on the ultraflat single-crystal Cu(111) substrate that was obtained by epitaxial growth of Cu(111) (i.e. Cu(111) film) on annealed c-plane sapphire. prepared on c-plane sapphire, Raman characterization has also been conducted to evaluate the AB-stacking structures after transferring the BLG samples onto SiO₂/Si substrates. Note that the Raman spectra were acquired every 1 mm on three cm-sized BLG samples grown on Cu(111)/Sapphire substrates (Supplementary Fig. 27). For all the spectra, the I_{2D}/I_G value is between 0.5 to 1 and the FWHM(2D) value is between 40 to 60 cm⁻¹, furthering confirming the successful preparation of AB-BLG (100% ratio) over the large area.’ (Page 17, 2nd paragraph in the updated supporting information)

3. Can the author make AB 100% BLG in 20 min? It is unclear 20 min is time for making a full BLG sheet or for growing AB 100% BLG. The authors’ claim is not very clear.

Reply: Thanks for the comments. 20 min is the duration for growing a continuous BLG film on the Cu foils substrates, rather than growing AB 100% BLG in our work. We feel sorry for the incurred misleading and have provide more detailed information to clarify it in the following.

First, with the aid of CO₂, continuous polycrystalline AB-BLG film with AB-stacking ratio of ~61% can be obtained on the commercial polycrystalline Cu foils across meter-sized region. To improve the ratio of AB-BLG and reduce the non-AB ratio, two kinds of single-crystal Cu(111) substrates were used, including the cm-sized Cu(111) foils which derived from the high-temperature annealing of the commercially available Cu foil (i.e. Cu(111) foil) (*ACS Nano* 2022, 16, 285) and the ultraflat single-crystal Cu(111) substrate that was obtained by epitaxial growth of Cu(111) (i.e. Cu(111) film) on annealed c-plane sapphire (*ACS Nano* 2017, 11, 12337). For Cu(111) foils, continuous BLG film can be grown within 20 min and the ratio of AB-BLG is up to 96% and this ratio might be further increased to 100% by further improving the purity (*Adv. Sci.* 2022, 9, 2200217) and surface smoothness (*Adv. Mater.* 2015, 27, 1376) of Cu(111) foils.

In this work, to further improve the AB-BLG ratio to 100%, the ultraflat single-crystal Cu(111) substrate on c-plane sapphire was used. The recipe we used for growing 100% AB-BLG films is as following: ‘After 30 min high-temperature annealing at 1,020 °C using 1,000 sccm Ar and 50 sccm H₂, 200 sccm diluted CH₄ (0.1%, in Ar) is introduced for 60 min to acquire a

continuous MLG film, followed by flowing 30 sccm CO₂ for 30 min to prepare a continuous BLG'. The growth of this bilayer graphene films requires 90 min, because the graphene growth on the 500 nm thick Cu film needs to be conducted under atmosphere pressure to minimize Cu sublimation, and CO₂ needs to be introduced after the formation of continuous MLG owing. Otherwise, no continuous graphene can be obtained owing to the increased etching ability of CO₂ which is caused by its decreased flow velocity and thus the longer staying time of CO₂ in the atmosphere-pressure CVD system. Since the amount of CH₄ used for growing MLG was set small to avoid the formation of multilayer graphene in this stage, a longer time was consumed in the growth of the graphene layer. According to our previous experiences in BLG growth on Cu foils (Supplementary Fig. 3), it is also possible to further decrease the growth time of 100% AB-BLG on Cu(111) film by optimizing the CVD parameters, such as increasing the CH₄ concentration in the Ar gas and fine tuning the H₂, CH₄, and CO₂ ratio. This have been listed as one of our future research plans.

Overall, in this work, we mainly focus on the fast synthesis of BLG on large-area commercial polycrystalline Cu foils. Moreover, the compatibility of this CO₂-assisted process with the preparation of 100% AB-BLG on Cu(111) substrates have also been proved.

To make our work more informative and avoid the further misunderstanding, the growth parameters utilized for 100% AB-BLG growth on Cu(111)/sapphire substrates have been added in the updated Method section (Page 12, lines 278-281 in the updated main text). Above discussions have also been added in the updated supporting information (Page 15, 1st paragraph in the updated supporting information) with four new citations as Refers. 26-29.

4. Line 85: The authors claim “meter-sized BLG films high uniformity”. However, when I carefully check Fig. 1c, there are at least two 3L graphene grains at the left bottom corner and right edge of Fig. 1c. This indicates a possibility that there can be a number of 3L grains in the BLG film, in contrast to the authors’ claim (“meter-sized BLG films high uniformity”).

Reply: Thanks for the valuable comment. The claim of ‘meter-sized BLG films high uniformity’ is supported by the statistical results of bilayer coverage in Fig. 1g (Fig. R29i). Compared with the reported results, bilayer coverage no less than 94% with small ratio of multilayer graphene (<5%) in meter-sized region should be a great breakthrough towards mass production of BLG films on Cu (Fig. R29g). More OM images of the BLG in large area have

been supplied in the new supporting information as Supplementary Fig. 5 to support this (Fig. R30). Furthermore, to avoid misleading, OM image in Fig. 1c has been updated (Fig. R29c) and the statement of ‘high uniformity’ has been deleted in the updated main text. Instead, we quantitatively describe the ratio of bilayer graphene, as below ‘*Herein, we demonstrated the role of CO₂ in the formation of BLG and achieved the fast synthesis of continuous BLG within 20 min on the commercial polycrystalline Cu foils, which is compatible with the production of meter-sized BLG films with bilayer coverage no less than 94%.*’ (Page 4 and Page 5, lines 98-101 in the new main text)

Figure R29. Fast synthesis of large-area BLG film with the assistance of CO₂. a-c OM images of the transferred graphene after 1 min (a), 10 min (b), and 20 min (c) growth on commercial polycrystalline Cu foils. d Relationship between the graphene bilayer coverage and the flow rate of CO₂. Inset: Typical OM images of the graphene grown using 5 sccm (top left) and 20 sccm (bottom right) CO₂. e Relationship between the graphene growth time and its bilayer coverage when 30 sccm CO₂ was introduced to grow graphene for 20 min. f Relationship between the graphene growth time and its bilayer coverage when 30 sccm CO₂ was utilized only in the first 10 min. Inset: OM image of the synthesized graphene after flowing H₂ and CH₄ for 90 min but only flowing CO₂ in the first 10 min. g Relationship between the graphene growth time and its bilayer coverage, showing the advantage of our CO₂-assisted strategy (red) in comparison with previous works (blue). h Photographs of the large-area

BLG films grown on eight pieces of commercial Cu foils in one batch. **i** Statistic of the graphene bilayer coverage of the graphene samples in **(h)** (red) and the graphene sample grown without CO₂ (green).

Figure R30. OM images of BLG acquired every 1 cm distance.

To grow TLG on Cu, abundant defective structures in the bottom graphene layer of the pre-formed BLG are required to allow the carbon species diffuse into the interface between the bottom graphene layer and the Cu substrate underneath. However, with the increased graphene thickness, it is more difficult for CO₂ to etch graphene to create such defects, as supported by both of our experimental results and first-principle simulations, together with the previous reports, which are discussed in detail below. According to our experimental results, the nucleation density, growth rate and coverage of TLG graphene are much lower than those of BLG (Fig. R31), which enables the large-area fast synthesis of BLG films with negligible TLG coverage (<5%) on Cu substrates after systematically optimizing CVD parameters in this work (Fig. R30).

Figure R31. OM image of the graphene film transferred on a SiO₂/Si substrate, which is composed of MLG, BLG, TLG and few-layer graphene.

First principle simulations have also been conducted by constructing the MLG, BLG and TLG structures on Cu(111) surface to quantitatively investigate the difference of CO₂ etching ability on multilayer graphene. We know that the etching rate of graphene (V_g) at a specific temperature depends on the etching difficulty of single carbon atoms in graphene (V_c) and the flux of etching gas (F_{CO_2}), scilicet as: $V_g \propto V_c * F_{CO_2}$. Since the chemical environment and coordination structure of all C atoms in the perfect multilayer graphene are the same, we believe that the etching rate of C atoms in each graphene layer is the same and can be set as constant under specific CVD conditions. Therefore, the etching difficulty of graphene with different layers only depends on the effective F_{CO_2} contacting it. Further on, the F_{CO_2} can be calculated by the concentration of CO₂ in contact with graphene of different layers ($\rho = e^{-\frac{E_{ad}}{k_B * T}}$) and the corresponding effective exposure area (S), which can be written as:

$$F_{CO_2} = S * e^{-\frac{E_{ad}}{k_B * T}}$$

where k_B and T are Boltzmann constant and experimental temperature of 1,300 K, respectively. As illustrated in Fig. R32a-c, the adsorption energy (E_{ad}) of CO₂ molecules on the MLG and BLG surfaces were calculated to be -0.07 eV and -1.44 eV, respectively, which indicates the reduced concentration for CO₂ to be captured by BLG in comparison with the MLG. At the same time, the exposure area (S) of the bottom graphene layer to CO₂ also decreases with the layer number. Bringing into the numerical calculation, we can obtain the F_{CO_2} values (Fig. R32d). It is clearly observed that the etching flux of graphene decreases significantly with its layer number. Moreover, since the real defect density of BLG is much lower than that estimated from the calculation model we built here (Fig. R32b), the formation of TLG with the aid of CO₂ is thus even much more difficult.

Figure R32. Theoretical estimation of CO₂ flux on MLG, BLG and TLG. **a** Configurations of CO₂ molecules adsorbed on the MLG (**a**), BLG (**b**) surfaces. **c** Adsorption energy of CO₂ molecules on the MLG (green), and BLG (red). **d** Comparison of the etching flux of CO₂ on MLG (green) and BLG (red) surfaces as a function of temperature.

Our finding is in good agreement with the calculation results by Wu *et al.* (*J. Phys. Chem. C* 2014, 118, 6201). They also reported that the energy barrier for C atoms to penetrate BLG (2.94 eV) is much higher than that to penetrate MLG (1.48 eV), indicating that growing a third layer graphene is much more difficult than growing BLG and thus it is highly possible to achieve the layer control during graphene growth on Cu.

To clearly explain the mechanism of CO₂-assisted BLG growth on Cu, especially in layer number control, Fig. R32 has been added in the updated version as Supplementary Fig. 7, together with above discussions (Page 6 and Page 7, lines 1-11 in the updated supporting information).

5. If CO₂ makes holes for the entrance of carbon feedstock, there can be many defects in top graphene layer. This may result in a strong Raman D band. However, the Raman spectra in Fig. 3g shows negligible D band. Please explain this.

Reply: Thanks for the valuable comment. Negligible D band in the Raman spectra of our BLG is attributed to the rapid repairing of graphene defects. Experimentally, the CO₂ gas is immediately turned off after the high-temperature growth stage, while CH₄ and H₂ gases still continuously flow into the CVD system, which would enable the continuous repairing of the as-produced defects. Therefore, during the cooling stage from 1,020 °C to 500 °C (*Appl. Phys. Lett.* 2013, 102, 103107; *ACS Nano* 2015, 28, 3428), which takes >10 min, the decomposition of CH₄ could provide enough carbon species for the healing of graphene defect created by CO₂.

Note that during the BLG growth process, once CO₂ was turned off, the bilayer coverage cannot increase even continuously supplying CH₄ and H₂ for extra 80 min (Fig. R33), which also proves that the defects in the top graphene layer can repair rapidly. Furthermore, Geng *et al.* (*ACS Cent. Sci.* 2022, 8, 394) also reported that high-quality BLG can be synthesized when CO₂ was used to grow large BLG domains, which agrees well with our experimental results.

Figure R33. Rapid healing of defective graphene at high temperature on Cu substrate. a Relationship between graphene growth time and its bilayer coverage when 30 sccm CO₂ was introduced for graphene growth for 20 min. **b** Relationship between graphene growth time and its bilayer coverage when 30 sccm CO₂ was utilized only in the first 10 min. Inset: OM image of the synthesized graphene after flowing H₂ and CH₄ for 90 min but only flowing CO₂ in the first 10 min.

We have conducted first-principle calculations using the model of BLG/Cu to explore the elementary reaction of C active species (dominated by CH particles) from free state to splicing to the upper and lower layers of graphene respectively, and evaluate the self-healing mechanism of BLG process according to the structural evolution and energy profiles (Fig. R34). In general, the CH species used for graphene growth can be from the gas phase (Fig. R34b) or the catalyst surface (Fig. R34c). For both conditions, we found an obvious energy decrease of > 7.4 eV after the CH species reaches a stable adsorption state, indicating that the carbon species can be easily captured by graphene defect edges and thus supplies the essential C atoms to form perfect graphene. Moreover, the splicing energy of CH at the edge of the upper and lower graphene defects is almost equal, which means that the defect healing of the upper graphene is almost synchronous with the growth of the lower graphene. After that, rapid self-healing of the defective graphene at high temperature on Cu substrates will happen, as reported by Wang *et al.* (*J. Am. Chem. Soc.* 2013, 135, 4476). Therefore, high-quality BLG with negligible D band observed in Raman can be obtained in our work.

Figure R34. Defect-healing simulation of BLG on Cu substrate with CH species. **a** Model configuration of the defective BLG on Cu(111) substrate. **b,c** Potential energy profiles and corresponding structural evolutions for BLG defect edges capturing CH species from the gas phase (**b**) and Cu substrate (**c**).

To make this manuscript more informative, Fig. R34 has been added as Supplementary Fig. 15, together with relevant discussions (Page 10, last 4 lines and Page 7, lines 1-11 in the updated supporting information), together with 5 new citations as Refers. 7, 12-15.

The followings are more minor comments:

6. Fig. 1i: No experimental evidence is shown. OM images for every 1cm distance should be presented to prove Fig. 1i.

Reply: Great thanks for the valuable suggestion to strengthen our presentation. Following the reviewer's suggestion, OM images of the transferred BLG samples, which had been acquired every 1 cm distance, have been included in the revised supporting information as Supplementary Fig. 8 (Fig. R35) to prove Fig. 1i, 30, together with corresponding discussions, as follows:

‘Supplementary Note 4. Mass production of BLG on commercial Cu foils with the aid of CO₂ Batch-to-batch production of large-area BLG film was conducted using the commercial polycrystalline Cu foils without special pre-treatment. To evaluate the bilayer coverage, OM images of the transferred BLG was acquired every 1 cm (Supplementary Fig. 8). Note that the

areal ratio of BLG was confirmed based on its contrast difference from that of MLG or few-layer graphene and for all measured positions, >94% coverage of BLG films was observed, indicating that the self-limited growth of graphene on Cu has been broken after introducing CO₂. (Page 7, lines 13-21 in the updated supporting information)

Figure R35. OM images of BLG acquired every 1 cm distance to prove Fig. 1i.

7. Fig. 3g: All five Raman spectra looks quite similar. This makes me feel very strange, because there should be some fluctuations in 2D band shape and intensity. Please indicate where you measured each Raman by using optical images.

Reply: Thank you for the comment. Following the reviewer's suggestion, the positions where the five Raman spectra were acquired have been marked in the OM image (Fig. R36a). Note that to collect these five spectra, a line mapping was conducted along the white dashed line by acquiring one Raman spectrum every 50 μm. However, the Raman peak intensity decreases owing to the focus change during the mapping measurement (Fig. R36b). Therefore, to minimize this effect, in Fig. 3d of the previously submitted manuscript, we displayed these five

Raman spectra after normalizing the G band intensity (Fig. R36c). We feel sorry for not mentioning this in the originally submitted version. At the same time, there are slight differences in peak position, full width at half maximum and intensity ratio of 2D and G bands for these five Raman spectra (Fig. R36d), which should be caused by the transfer step.

Figure R36. Raman characterization of the AB-BLG film in large area. a OM image of the BLG for Raman characterization. **b** Raman spectra collected on the five positions marked in (a), which were acquired by line scanning. **c** Raman spectra after normalization using G band intensity based on (b). **d** Magnified G and 2D bands of the five Raman spectra in (c).

Following the reviewer's suggestion, we have added the OM image as Supplementary Fig. 26a and marked the positions for both Raman spectra and mapping measurement on it, together with relevant discussions in the revised version as below.

'Note that the five Raman spectra displayed in Fig. 4b were firstly acquired in the marked positions in Supplementary Fig. 26a through a line mapping with the step of 50 μm and then normalized using the intensity of G bands.' (Page 16, last 2-4 lines and Page 3, 1st line in the updated supporting information)

'b Raman spectra of the AB-BLG transferred onto a SiO₂/Si substrate acquired via a line scanning with the step of 50 μm by normalizing the G band intensity.' (Page 21, lines 513-515 in the updated main text)

8. *Fig. 4a inset: Please describe how you measured the transmittance mapping. Because a standard UV-vis spectrometer measures only at one specific point.*

Reply: Thank you for the comment. To conduct the transmittance mapping of the MLG and BLG samples in large area, we firstly transferred them onto quartz substrates and then manually moved the samples every 1 cm in both horizontal and vertical directions to acquire their transmittance at 550 nm wavelength using a commercial UV-vis spectrometer. For each sample, 9 positions were measured to evaluate its uniformity in $\sim 3 \times 3$ cm²-sized region. Photograph of the graphene/quartz samples has been added in the supporting information with the measured regions clearly marked as Supplementary Fig. 20 (Fig. R37), together with above discussions (Page 14, lines 1-5 in the updated supporting information). In addition, one sentence has been added in the new main text as below '*and the transmittance mapping is conducted by manually adjusting the sample positions.*' (Page 13, lines 303-304 in the new main text)

Figure R37. Photograph of the MLG and BLG transferred to quartz substrates for transmittance measurement in large area.

9. *Fig. 4b: The sheet resistance is much lower than that widely reported for BLG (300-1000 Ohm/sq). Please describe how you measured the sheet resistance including the electrode size and spacing. Did the authors experience the BLG film breakage using CDE RsMap 178? This is because this equipment probably uses four pins directly attaching to a graphene sheet.*

Reply: Thank you for the comment. As mentioned in the method section, after the polymethyl methacrylate (PMMA)-assisted transfer of the MLG and BLG films onto SiO₂/Si substrates, their sheet resistance is directly measured using CDE RsMap 178 instead of by fabricating microdevices. Distances between the 4 pins are 1 mm. Following the reviewer's suggestion,

we found that after repetitive measurement of the same region, no obvious BLG film breakage was observed, as confirmed by both the mapping and histogram statistical results (Fig. R38). This can be attributed to the proper setting of the probe height to lift when approaching and contacting the graphene films on SiO₂/Si substrates.

There are several reasons for the observed reduced sheet resistance of BLG in comparison with MLG. Firstly, according to the equation $\sigma = n * e * \mu$, the electrical conductivity (σ) of graphene is decided by both the carrier mobility (μ) and carrier concentration (n). For BLG, even though the carrier mobility is decreased in comparison with MLG, the increased number of conduction channels would contribute to the increased carrier concentration and thus increases the electrical conductivity (*J. Supercond. Nov. Magn.* 2017, 30, 1263). The increased electrical conductance of BLG has been widely investigated and was supported by both theoretical simulation and experimental results (*Energy Environ. Sci.* 2013, 6, 108; *Nano Lett.* 2009, 9, 8, 2973; *Nat. Nanotechnol.* 2010, 5, 574; *Adv. Mater.* 2011, 23, 1514; *Nano Lett.* 2009, 9, 12). Secondly, our BLG is high quality and has low defect density, as indicated by the negligible D band in Raman spectra, which also contributes to the improved electrical conductivity (*Acta Phys.-Chim. Sin* 2019, 35, 1000). Thirdly, the BLG used for sheet resistance measurement is grown on the Cu(100)-dominant polycrystalline Cu and is composed of ~ 61% AB-BLG and ~39% tBLG by area; the presence of tBLG areas with high twist angles also contributes to decreasing sheet resistance of the BLG film, as reported by Yuji Araki *et al.* (*ACS Nano* 2022, 16, 14075). Finally, the enhanced mechanical property of BLG and the optimized transfer process can reduce the possibility of crack formation during transfer compared with that of MLG, which in turn benefits the reliable measurement of sheet resistance values in large area.

To make our manuscript more informative, we have added above discussions in the updated version (Page 14, last 2 paragraphs in the updated supporting information), together with eight new citations as Refers. 18-25.

Figure R38. Sheet resistance measurement of BLG. **a** Sheet resistance mapping result of the BLG transferred onto SiO₂/Si substrates with step of ~0.5 mm. **b** Statistical histograms of the sheet resistance of the BLG based on (a). **c** Sheet resistance mapping result of the BLG transferred onto SiO₂/Si substrates with the step of ~1 mm. **d** Statistical histograms of the sheet resistance of the BLG based on (c).

10. Fig. 4d: There is no transfer curves for the hBN/BLG/hBN device with different top and bottom gate voltages. Generally, researchers present transfer curves when they discuss the band gap opening in BLG samples. Also, the temperature of the device measurement should be provided.

Reply: Thanks for your valuable comment. For the electrical transport measurement, graphene grown on a Cu (111) foil, which was composed of isolated AB-BLG domains and continuous MLG film was utilized (Fig. R39a). After Raman characterization across the whole domain (Fig. R39b), the BLG domain was picked up by a hBN nanoflake to fabricate the hBN/BLG/hBN sandwiched structure (Fig. R39c). Note that for all the Raman spectra, the I_{2D}/I_G is between 0.5 to 1 and the FWHM(2D) value is between 40 to 60 cm⁻¹, confirming that this BLG domain is AB-stacking structure (*Nat. Commun.* 2019, 10, 2809).

Figure R39. OM and Raman characterization of the AB-BLG domain used to fabricate the dual-gate device. a OM image of the AB-BLG domain transferred on a SiO₂/Si substrate. **b** Raman spectra acquired on four different positions marked in (a). **c** OM image of the hBN/BLG/hBN sandwiched structure.

Our previous measurement was conducted in a glovebox filled with Ar atmosphere at 300 K, on which condition the impact of thermal fluctuation and interface contamination cannot be ignored. In specific, impacts on the electrical transport measurement of the BLG from both the polymer residues atop the graphene, introduced during the wet transfer, and the bubbles formed during the stacking, cannot be avoided, since they severely impact the interface quality between hBN and graphene, especially considering that the density and size of bubbles in the BLG region are much larger than those in the MLG regions (Fig. R5c). To further reduce the impact from the measurement environment and surface/interface contaminations, we have loaded the same device into a vacuum chamber ($\sim 10^{-7}$ Torr) and measured it again at 290 K. This time the opening of band gap has been observed more clearly (Figs. R40 and 41). In detail, when sweeping the bottom-gate voltage from -15 V to 50 V and sweeping the top-gate voltage (4 V to -15 V), the resistance reached maximum at highest displacement field region (the top-left and bottom-right), confirming the tunability of bandgap with perpendicular dipole electric field (Fig. R40d). The transfer curves of the AB-BLG when sweeping the bottom gate voltage is given in Fig. R41, in which the maximum resistance attained at the charge neutrality point increases with increasing top gate voltage in both the positive and negative directions, further confirming the opening of the band gap.

Figure R40. Synthesis of AB-BLG on Cu(111) substrates. **a** Distribution of twist angles based on SAED patterns of the BLG grown on the ultraflat single-crystal Cu(111) substrate that was obtained by epitaxial growth of Cu(111) (i.e. Cu(111) film) on annealed c-plane sapphire. Inset: Statistical results of the stacking order (AB stacking or non-AB stacking) (centre) and STEM image of the AB-BLG with atomic resolution (top right). **b** Raman spectra of the AB-BLG transferred onto a SiO₂/Si substrate acquired via a line scanning with the step of 50 μm by normalizing the G band intensity. **c** Raman 2D band of the AB-BLG, which is fitted by four Lorentzian peaks. **d** Two-dimensional plot of the ρ_{xx} as functions of both V_{tg} and V_{bg} of a dual-gate AB-BLG device. **e** Photocurrent distribution in the two graphene/electrode junctions. Inset: OM image of the graphene FET device. Scalebar: 5 μm. **f** Photocurrent mapping result of the graphene device, in which the MLG is on the left part and the BLG is on the right part.

Figure R41. Transfer curves of the dual gate AB-BLG Hall bar device with varied V_{tg} and V_{bg} .

According to the reviewer's concern, the mapping result has been updated in Fig. 4d (Fig. R40). Raman and OM characterization of the AB-BLG (Fig. R39) and transfer curves of the device with varied V_{tg} and V_{bg} (Fig. R41) have also been added in the updated supporting information

as Supplementary Figs. 28 and 30, accompanied by relevant discussions (Page 17, last 4-10 lines and Page 18, 2nd paragraph in the updated supporting information). Meanwhile, relevant discussion in the Method part have also been revised, as below.

‘The carrier mobility was measured at 300 K in a glovebox filled with Ar and the gate-dependent transfer curves were measured at 290 K under vacuum ($\sim 10^{-7}$ torr).’ (Page 14, lines 310-312 in the updated main text)

11. Supplementary Fig. 14: It is unclear for me where and how you measured “Representative HRTEM images”. More detailed explanation is necessary. Ideally, these images should be collected at different positions far apart each other.

Reply: Thank reviewer #3 for the valuable comment. The HRTEM images shown in Supplementary Fig. 14 are acquired using the BLG film grown on a Cu(100)-dominated polycrystalline Cu foil. After acquiring the SAED patterns in Supplementary Fig. 18, HRTEM images from positions far away from each other (adjacent distances $> 500 \mu\text{m}$) are acquired across the 3 mm-sized TEM grid to evaluate crystallinity of the BLG in large area. As a result, in addition to the three AB-BLG regions, tBLG with different twist angles are also characterized using HRTEM. To avoid possible misunderstanding, we have deleted the statement ‘representative’ in our manuscript and added more experimental details, as below.

‘For the same BLG film grown on Cu(100)-dominated polycrystalline Cu foils, after acquiring SAED patterns, HRTEM images from positions far away from each other (adjacent distance $> 500 \mu\text{m}$) were acquired across the 3 mm-sized TEM grid, from which the point defects were seldom observed, further implying high crystallinity of the BLG. Note that in addition to the three AB-BLG regions, six tBLG regions with different twist angles were also characterized using HRTEM (Supplementary Fig. 18).’ (Page 12, lines 14-19 in the updated supporting information).

12. Supplementary Fig. 17: Generally, the carrier mobility of BLG is low, due to its parabolic band structure. However, the authors observed very high carrier mobility over $5000 \text{ cm}^2/\text{Vs}$. Why did they obtain such high mobility for AB BLG?

Reply: Thanks for the valuable comments from reviewer 3#. We need to politely point out that

even though with its parabolic band structure the carrier mobility of BLG is lower than that of MLG, the room temperature carrier mobility of over $5,000 \text{ cm}^2\text{V}^{-1}\text{s}^{-1}$ is not surprisingly high for high-quality BLG. Experimentally, the room-temperature carrier mobility of CVD AB-BLG on SiO_2/Si substrates has been widely reported to be higher than $2,000 \text{ cm}^2\text{V}^{-1}\text{s}^{-1}$ (*Adv. Funct. Mater.* 2015, 25, 3666; *ACS Nano* 2012, 6, 8241; *Nat. Nanotechnol.* 2020, 15, 289; *Nat. Commun.* 2019, 10, 1). Moreover, the carrier mobility values can be further increased by using hBN as dielectrics to minimize the scattering effect from underlying SiO_2/Si substrates (*Science* 2013, 1, 614.). In our work, since the BLG has high quality, as confirmed by Raman and HRTEM observation, and mechanical exfoliated hBN flakes are utilized as both the top and bottom gate dielectrics to minimize the impact from external environment, we observe a high room-temperature carrier mobility $> 5,000 \text{ cm}^2\text{V}^{-1}\text{s}^{-1}$ for the AB-BLG sample.

13. Fig, 1g: I think the references correspond to Supplementary Table 1. It is better for general readers to mention that the original references are cited in Supplementary Table 1.

Reply: Thanks for the beneficial comments. Following the reviewer's suggestion, we have added one sentence in the main text '*Note that the original references are also listed in Supplementary Table 1 with more details^{9,10,17,21,22,32-52.}*' (Page 6, lines 136-137 in the new main text) and one sentence in the supporting information '*Note that for clear comparison, only those with BLG growth time shorter than 200 min are plotted in Fig. 1g, while the others are only listed in the Supplementary Table 1.*' (Page 34, lines 3-4 in the updated supporting information) to point out the relation between Fig. 1g and Supplementary Table 1. Meanwhile, we have also added the reference numbers in Fig. 1g (Fig. R42) to increase the readability of this manuscript.

Figure R42. Fast synthesis of large-area BLG film with the assistance of CO₂. a-c OM images of the transferred graphene after 1 min (a), 10 min (b), and 20 min (c) growth on commercial polycrystalline Cu foils. d Relationship between the graphene bilayer coverage and the flow rate of CO₂. Inset: Typical OM images of the graphene grown using 5 sccm (top left) and 20 sccm (bottom right) CO₂. e Relationship between the graphene growth time and its bilayer coverage when 30 sccm CO₂ was introduced to grow graphene for 20 min. f Relationship between the graphene growth time and its bilayer coverage when 30 sccm CO₂ was utilized only in the first 10 min. Inset: OM image of the synthesized graphene after flowing H₂ and CH₄ for 90 min but only flowing CO₂ in the first 10 min. g Relationship between the graphene growth time and its bilayer coverage, showing the advantage of our CO₂-assisted strategy (red) in comparison with previous works (blue). h Photographs of the large-area BLG films grown on eight pieces of commercial Cu foils in one batch. i Statistic of the graphene bilayer coverage of the graphene samples in (h) (red) and the graphene sample grown without CO₂ (green).

REVIEWER COMMENTS

Reviewer #1 (Remarks to the Author):

I appreciate very much the efforts of authors in providing tremendous additional experiment and calculation data to address reviewer's comment/concern. Most of my concerns and comments have been clearly resolved. I think the revised manuscript have become much stronger and are almost ready for publication.

Reviewer #2 (Remarks to the Author):

In the previous review, my main concern was clarity of the role of CO₂ in BLG growth, discussion of the research trend of trace CO₂ supply during growth process and distinguishing this work from other reported CO₂ assisted BLG growth works (such as Gong et al ACS Cent. Sci. 2022, 8, 394–401 and Seekwaew et al Membranes 2022, 12, 796.) The authors have carefully provided point-by-point explanation to the concerns raised and have modified the manuscript accordingly. Therefore, I recommend that this manuscript could be published in Nature Communication without further revision.

Reviewer #3 (Remarks to the Author):

The revised version of the manuscript seems much improved than the original. I appreciate the effort of the authors to reply to the issues raised by the referees. There are still some issues that seem worth being revised before recommending the publication of the manuscript (detailed below). Other than that, while I appreciate the inclusion of the references concerning the use of CO₂, I cannot agree with the reasoning from the authors about why none of them were included in the original draft (literally "We have carefully read the mentioned references, and we would like to point out that our work is different from the previous papers regarding the growth mechanism and quality of as-obtained BLG"). The fact that the results from a work being sent for publication differs/improves from earlier ones should be the norm. However, that does not mean that previous attempts should be overlooked. Reading the original draft seemed like the authors were the first that came with the idea of using CO₂ during the CVD, which is not true. Although difficult to prove, the fact that two of the referees were aware of these previous works makes it difficult to think that none of the authors knew about them and/or were inspired by them for the present work. This lack of acknowledgements does not seem to be an isolated case for some of the authors of the present work (as an example, it personally seems a big omission not to cite doi: 10.1021/nn100459u in the paper doi: 10.1038/s41467-020-14359-0).

The following are some issues that I would like the authors consider addressing:

- 1) It seems strange that in the original draft, the authors did not seem to find an optical image of purely 2L graphene. Instead, the original figure included 3L areas, which reading the manuscript and the claims of high 2L coverage seems unlikely. I am thus wondering why that figure was included in the first place, and would like to know how large are the areas of pure 2L that can be found. Sadly, the quality of the supplementary figure 8 is not that good.
- 2) Regarding the new Fig. 1c, the image seems to be noisier than the other optical images. Is there any explanation for this?
- 3) One of the original questions (number 9), was about the sheet resistance of the current BLG being lower than values reported for BLG. However, the authors' reply was about comparing the sheet resistance of BLG and MLG. So, it is still not clear why the current BLG sheet resistance is that low.
- 4) From the electrical characterization, it seems that the graphene is p-doped, but nothing is mentioned in the manuscript. Is this the case? If so, it would be interesting to know the origin of this doping, and whether this is seen in other measurements (maybe Raman). Can this doping be also the cause for the low sheet resistance?
- 5) About the new measurement to determine the opening of the band gap of BLG, the authors claim in a response to one of the other referees that "the impact of thermal fluctuation and interface contamination cannot be ignored" for the original measurements at 300 K. However, the new measurements have been conducted at 290 K, which does not seem to be difference large

enough to account for the observed differences. The measurement atmosphere was also changed, from ambient pressure Ar to low pressure. However, the hBN should be protecting the graphene channel, and hence the differences should in principle be not that large. Can the authors comment on these issues?

6) Related to the previous comment, the noise in the new measurements (Fig. 4d) seems similar to the noise of the previous one (for example the noise for the region of low top gate and large bottom gate). Is there any reason or explanation for this?

7) Can the authors check if Supplementary Fig. 29d should be for the conductance?

8) In general, the quality of the images of the supporting information is low, rendering them useless (see the previous comment about Figure 8 of the supporting information)

Reviewer #1 (Remarks to the Author):

I appreciate very much the efforts of authors in providing tremendous additional experiment and calculation data to address reviewer's comment/concern. Most of my concerns and comments have been clearly resolved. I think the revised manuscript have become much stronger and are almost ready for publication.

Reply: We sincerely thank the reviewer #1 for the great effort in reviewing our manuscript and appreciate very much for the explicit recommendation of publication.

Reviewer #2 (Remarks to the Author):

In the previous review, my main concern was clarity of the role of CO₂ in BLG growth, discussion of the research trend of trace CO₂ supply during growth process and distinguishing this work from other reported CO₂ assisted BLG growth works (such as Gong et al ACS Cent. Sci. 2022, 8, 394–401 and Seekwaew et al Membranes 2022, 12, 796.) The authors have carefully provided point-by point explanation to the concerns raised and have modified the manuscript accordingly. Therefore, I recommend that this manuscript could be published in Nature Communication without further revision.

Reply: We appreciate very much for the positive comments from the reviewer # 2 and for the explicit recommendation of publication.

Reviewer #3 (Remarks to the Author):

The revised version of the manuscript seems much improved than the original. I appreciate the effort of the authors to reply to the issues raised by the referees. There are still some issues that seem worth being revised before recommending the publication of the manuscript (detailed below). Other than that, while I appreciate the inclusion of the references concerning the use of CO₂, I cannot agree with the reasoning from the authors about why none of them were included in the original draft (literally “We have carefully read the mentioned references, and we would like to point out that our work is different from the previous papers regarding the growth mechanism and quality of as-obtained BLG”). The fact that the results from a work being sent for publication differs/improves from earlier ones should be the norm. However, that does not mean that previous attempts should be overlooked. Reading the original draft seemed like the authors were the first that came with the idea of using CO₂ during the CVD, which is not true. Although difficult to prove, the fact that two of the referees were aware of these previous works makes it difficult to think that none of the authors knew about them and/or were inspired by them for the present work. This lack of acknowledgements does not seem to be an isolated case for some of the authors of the present work (as an example, it personally seems a big omission not to cite doi: 10.1021/nn100459u in the paper doi: 10.1038/s41467-020-14359-0).

Reply: We appreciate very much for the great efforts in reviewing our manuscript and for the positive comments on the revised version of this work from Reviewer #3. We sincerely hope that the inclusion of the references concerning the use of CO₂ in our revised version helps to avoid misleading the readers to overlook previous attempts in this fields and we will pay much more attention in paper citations in the future to avoid such issues. Following the valuable comments and suggestions from the reviewer #3, we have made point-to-point response and revisions.

The following are some issues that I would like the authors consider addressing:

1) It seems strange that in the original draft, the authors did not seem to find an optical image of purely 2L graphene. Instead, the original figure included 3L areas, which reading the manuscript and the claims of high 2L coverage seems unlikely. I am thus wondering why that figure was included in the first place, and would like to know how large are the areas of pure 2L that can be found. Sadly, the quality of the supplementary figure 8 is not that good.

Reply: Thanks for the comments. In the original submitted version, an optical microscope (OM) image that includes a small area of trilayer graphene was put in Fig. 1c to correspond to the statistical results about the bilayer coverage (95%-100%) in Fig. 1i. In the updated version we submitted last time, following the kind suggestions from Reviewers, to avoid misleading, we have updated Fig. 1c using an OM image with only bilayer graphene (BLG).

In this work, we have obtained pure BLG with size no smaller than millimetre, as confirmed by the OM images acquired using 5X objective (Fig. R1). As shown in Fig. R1a, the contrast difference of bare SiO₂/Si, monolayer graphene (MLG) and BLG is easy to distinguish in the graphene breakage regions, while in Fig. R1b, for the intact BLG film, a uniform contrast is observed in large area ($\sim 2 * 3 \text{ mm}^2$).

OM images in Supplementary Fig. 8 were acquired using 10X objective under OM, which were inevitably compressed when we put 30 images together and output them as a new figure. The polymer residues introduced in the wet-transfer process also have some influences in the optical contrast. To improve the image quality, we have increased the resolution of this figure from 300 ppi to 1,200 ppi when outputting it from the Adobe Illustrator software (Fig. R2). After this operation, contrast difference caused by multilayer graphene, graphene breakage, and polymer residues is more visible. In addition, to provide more detailed information about the BLG films, we have taken OM images using 50X object after annealing six transferred BLG samples to minimize the effect of the polymer residues (Fig. R3).

Figure R1. OM images of the BLG in large area. a OM image of BLG with some breakage regions. **b** OM image of the intact BLG.

Figure R2. OM images of the BLG films acquired using 10X objective.

Figure R3. OM images of the BLG films on SiO₂/Si substrates collected using 50X objective.

Fig. R1 has been added in the updated supporting information as Supplementary Fig. 8 to provide more evidence for the successful synthesis of large-area BLG film. Fig. R2 has been used to replace the Supplementary Fig. 6 in the previous version as Supplementary Fig. 11 and Fig. R3 has been added as Supplementary Fig. 12 in the updated supporting information. The relevant discussions have also been added, as below.

‘As a result, large-area BLG film can be obtained (Supplementary Fig. 8). The contrast difference of bare SiO₂/Si substrate, MLG and BLG is clearly seen in Supplementary Fig. 8a, while for intact regions, a uniform contrast in at least mm-sized areas is observed (Supplementary Fig. 8b).’ (Page 5, last 5 lines in the updated supporting information)

‘To evaluate the bilayer coverage, OM images of the transferred BLG were acquired every 1 cm (Supplementary Fig. 11) using 10X objective under optical microscope, which cover sub-mm sized area for each image. To provide more details about the BLG, OM images were also taken using 50X objective after annealing the samples to decrease disturbance of the transfer-induced polymer residues (Supplementary Fig. 12).’ (Page 5, last 5 lines in the updated supporting information)

2) Regarding the new Fig. 1c, the image seems to be noisier than the other optical images. Is there any explanation for this?

Reply: Thanks for the comments and we feel sorry for causing this confusion. The new Fig. 1c was taken using

10X objective while Fig. 1a,b were taken using 50X objective, which makes Fig. 1c noisier than the other two images. To tackle this, we have updated Fig. 1c using another OM image that was acquired using 50X objective (Fig. R4). Fig. R4 has been used to replace Fig. 1 in the updated main text.

Figure R4. Fast synthesis of large-area BLG film with the assistance of CO₂. a-c OM images of the transferred graphene after 1 min (a), 10 min (b), and 20 min (c) growth on commercial polycrystalline Cu foils. d Relationship between the graphene bilayer coverage and the flow rate of CO₂. Inset: Typical OM images of the graphene grown using 5 sccm (top left) and 20 sccm (bottom right) CO₂. e Relationship between the graphene growth time and its bilayer coverage when 30 sccm CO₂ was introduced to grow graphene for 20 min. f Relationship between the graphene growth time and its bilayer coverage when 30 sccm CO₂ was utilized only in the first 10 min. Inset: OM image of the synthesized graphene after flowing H₂ and CH₄ for 90 min but only flowing CO₂ in the first 10 min. g Relationship between the graphene growth time and its bilayer coverage, showing the advantage of our CO₂-assisted strategy (red) in comparison with previous works (blue). h Photographs of the large-area BLG films grown on eight pieces of commercial Cu foils in one batch. i Statistic of the graphene bilayer coverage of the graphene samples in (h) (red) and the graphene sample grown without CO₂ (green).

3) One of the original questions (number 9), was about the sheet resistance of the current BLG being lower

than values reported for BLG. However, the authors' reply was about comparing the sheet resistance of BLG and MLG. So, it is still not clear why the current BLG sheet resistance is that low.

Reply: Thanks. There are several factors accounting for the low sheet resistance of our BLG. In addition to its high crystallinity after growth and negligible breakage after transfer, which have been pointed out in our previous response letter, the strong *p*-doping caused by the polymethyl methacrylate (PMMA)-assisted wet transfer also helps to reduce the sheet resistance of BLG (*J. Vac. Sci. Technol. B*, 2012, 30, 041213; *RSC Adv.* 2017, 7, 6943; *Appl. Phys. Lett.* 2011, 99, 122108; *Sci. Rep.* 2016, 6, 30210). To verify the *p*-doping of our transferred BLG on SiO₂/Si substrate, we have fabricated Hall bar devices (Fig. R5a) and then conducted electrical measurements using the four-probe method. After measuring transfer curve of the device by applying bottom gate voltage from -30 V to 60 V, no charge neutrality point was observed, indicating strong *p*-doping of the BLG after transfer. At the same time, we have observed low sheet resistance at zero gate voltage (Fig. R5b), which is comparable to the result shown in our Fig. 3e,f in the main text.

Figure R5. Electrical measurement of the BLG. a OM image of the BLG device. **b** Transfer curve of the BLG device.

Doping analysis was conducted based on the Raman characterization results of the transferred BLG by distracting the peak positions of G and 2D peaks (Fig. R6), that is, Pos(G) and Pos(2D), following the previous work (*Nat. Commun.* 2012, 3, 1024.). Note that for BLG, the point of zero doping and zero strain should be located at Pos(G) = 1582 cm⁻¹ and Pos(2D) = 2682 cm⁻¹ (*Phys. Rev. Lett.* 2008, 101, 136804; *Phys. Status Solidi B* 2014, 251, 2545), which is different from that of MLG. Moreover, the dashed line with a slope of 0.7 can only serve as a guidance rather than quantitatively calculating the doping level of BLG owing to the non-monotonic behaviour of the optical phonon anomaly in BLG (*Phys. Rev. Lett.* 2008, 101, 136804). Qualitatively, in combination with the electrical measurement results, the obvious blueshift of G peak still provides solid evidence for the strong *p*-doping in the BLG, according to the gate-tuneable Raman measurement results reported previously (*Phys. Rev. Lett.* 2008, 101, 136804; *Phys. Rev. B* 2009, 79, 155417; *Phys. Rev. Lett.* 2007, 98, 166802).

Figure R6. Doping analysis of the transferred BLG using Raman. **a** Raman spectrum of the BLG transferred on SiO₂/Si for sheet resistance measurement. **b** Statistical result showing the relationship between Pos(G) and Pos(2D), in which the dashed blue line corresponds to the strain-free regions with slope of 2.45 while the dashed green line corresponds to the doping-free regions for MLG, which is plotted as a guidance here.

Fig. R6 has been added in the updated supporting information, together with relevant discussions and two new papers cited as Refs. 26 and 27, as below.

‘Moreover, similar to previous reported results²⁶, p-doping effect was observed in our BLG samples after the PMMA-assisted wet transfer (Supplementary Fig. 25)²⁷, which also contributes to decreasing the sheet resistance of BLG films.’ (Page 15, lines 6-8 in the updated supporting information)

4) From the electrical characterization, it seems that the graphene is p-doped, but nothing is mentioned in the manuscript. Is this the case? If so, it would be interesting to know the origin of this doping, and whether this is seen in other measurements (maybe Raman). Can this doping be also the cause for the low sheet resistance?

Reply: Thanks. We agree with Reviewer 3# that the graphene encapsulated by hBN flakes is *p*-doped according to the electrical characterization. As we discussed above, this is mainly because of the PMMA residues introduced during the transfer process, which has been reported to have obvious *p*-doping effect on graphene (*J. Vac. Sci. Technol. B*, 2012, 30, 041213; *RSC Adv.* 2017, 7, 6943; *Appl. Phys. Lett.* 2011, 99, 122108). In specific, to fabricate the hBN/BLG/hBN sandwiched structure, BLG film was first transferred onto SiO₂/Si substrate with the aid of PMMA using a wet method, followed by PMMA removal using acetone, after which the BLG was picked up from SiO₂/Si using one piece of hBN nanoflake (top gate) and then put on top of the other piece of hBN nanoflake (bottom gate) (Fig. R7a). Unlike the exfoliated graphene flakes, even though the hBN-based heterostructures has a self-cleaning effect, the PMMA residues atop the BLG prepared using chemical vapor deposition are hard to be totally removed, as indicated by the OM image (Fig. R7b). We think that this is the main reason for the *p*-doping in our electrical measurement.

During our experiment, Raman characterization was conducted by collecting four Raman spectra in the edge of the BLG domain on the SiO₂/Si substrate (Fig. R8a). As shown in Fig. 8b, obvious blueshift of G peak was observed, indicating the *p*-doping effect on graphene. As we discussed above, statistical results in Fig. R6 also confirmed the *p*-doping of our BLG after transfer, which we believe is one of the causes for the low sheet resistance of BLG.

Figure R7. PMMA residues atop of BLG. **a** Schematic of the PMMA-assisted wet transfer process of BLG onto SiO₂/Si substrate. **b** OM image of the hBN/BLG/hBN sandwiched heterostructure.

Figure R8. Raman analysis of the transferred BLG used for device fabrication. **a** OM image of the BLG on SiO₂/Si. **b** Raman spectra acquired in four positions marked in (a). **c** Magnified Raman spectra showing the blueshift of G peaks.

Discussions about the *p*-doping in the electrical measurement has been added, together with a new paper cited as Ref. 26 in the updated supporting information, as below:

*“The polymer residues also result in the *p*-doping of our BLG device²⁶, even if it is encapsulated by hBN flakes.”*
(Page 18, last 3-4 lines in the updated supporting information)

5) About the new measurement to determine the opening of the band gap of BLG, the authors claim in a

response to one of the other referees that “the impact of thermal fluctuation and interface contamination cannot be ignored” for the original measurements at 300 K. However, the new measurements have been conducted at 290 K, which does not seem to be difference large enough to account for the observed differences. The measurement atmosphere was also changed, from ambient pressure Ar to low pressure. However, the hBN should be protecting the graphene channel, and hence the differences should in principle be not that large. Can the authors comment on these issues?

Reply: Thanks. The difference in our experimental results is not caused by the 10 K temperature differences. Instead, it is mainly because that we had kept the device in high vacuum ($\sim 10^{-7}$ Torr) for one week and then measured it in the same condition. Previous papers have widely reported the influence of atmosphere on the performance of graphene electronic devices, especially the positive effect of vacuum measurement (*Nano Lett.* 2009, 9, 5; *Small* 2015, 11, 1402; *Sci. Rep.* 2016, 6, 205050). Moreover, differences in transfer curves of the graphene devices based on hBN encapsulated sandwiched structures have also been observed when changing the measurement atmosphere from air to vacuum (10^{-3} to 10^{-4} torr), especially after storing the device in vacuum for long time (*Appl. Phys. Lett.* 2015, 106, 193501; *ACS Appl. Mater. Interfaces* 2016, 8, 3072). Such phenomena have inspired us to put the BLG device into vacuum for electrical measurement.

Unlike the mechanical exfoliated graphene nanoflakes encapsulated by hBN flakes, the BLG prepared via chemical vapor deposition has more surface contamination after PMMA-assisted wet transfer. These residues inevitably impact the interface between graphene and the top hBN nanoflake, resulting in *p*-doping effect and the failure to open bandgap under ambient pressure Ar. After being kept in vacuum for long time (one week), the bubbles and contaminations trapped in the top hBN/BLG interface can move or release because of the giant pressure difference (10^7 torr), contributing to the improved BLG device performance and the observation of bandgap opening.

We are sorry for the misleading in our previous statement about the ‘*the impact of thermal fluctuation and interface contamination cannot be ignored*’. It was intended to claim the reported significant role of temperature when measuring graphene devices (*Phys. Rev. Lett.* 2010, 105, 166601), rather than to explain the differences observed in our own BLG device.

To strengthen our manuscript, one more sentence has been added both in the updated main text and in the updated supporting information, as below,

‘after storage in high vacuum ($\sim 10^{-7}$ torr) for one week.’ (Page 15, line 7 in the main text)

‘after keeping it in high vacuum ($\sim 10^{-7}$ torr) for one week.’ (Page 18, last line and Page 19, first line in the supporting information)

6) Related to the previous comment, the noise in the new measurements (Fig. 4d) seems similar to the noise of the previous one (for example the noise for the region of low top gate and large bottom gate). Is there any reason or explanation for this?

Reply: Thanks for the valuable comment to improve our manuscript. There are several possible sources of noise in electrical measurements, which can be classified into external and internal ones. The external noise comes mainly from the improper setup of the measurement equipment and relevant parameters (e.g., compliance), and disturbances from surrounding environments, while the internal ones are mainly because of the large contact resistance or unclean interfaces of the heterostructure. For example, the disturbance in the $V_{tg} \sim 0$ V region can be attributed to the internal factor, that is, the existence of dipole in the surface of BLG caused by PMMA residues.

The noise in the region of low top gate and large bottom gate regions is due to the external reason. In specific, when sweeping the bottom gate voltage (V_{bg}) from -15 V to 50 V and sweeping the top gate voltage (V_{tg}) from 4 V to -15 V, we had set the current compliance in source meter for the top gate as 10 nA to protect it from breakdown. However, the current reached 10 nA when V_{tg} increased to 13 V, resulting in the failure to apply V_{tg} higher than 13 V. That is, rather than following the values we set, the real V_{tg} applied to the BLG device was not stable but jumped to much smaller values, ranging from 9 - 13 V, resulting in two resistance values for one group of (V_{tg} , V_{bg}). However, we did not notice this phenomenon before the reminding from Reviewer #3. As a result, when we copied all data to plot a mapping image using Origin software, these noisy signals appeared in the bottom right region of Fig. 4d. When we measured the BLG device under high vacuum, we utilized same setup (e.g., range and step) for both V_{tg} and V_{bg} . After getting the new result, we kept using previous V_{tg} , V_{bg} values and only updated the resistance values in the previous Origin file to use the same format of the mapping image. Therefore, the jumping of V_{tg} in the previous measurement caused the reappearance of noise in the same positions in the mapping result in last version.

After masking this part of data, we have removed the noisy points and improved the image quality (Fig. R9). Fig. R9 has been used to replace Fig. 4 in the updated main text.

Figure R9. Synthesis of AB-BLG on Cu(111) substrates. **a** Distribution of twist angles based on SAED patterns of the BLG grown on the ultraflat single-crystal Cu(111) substrate that was obtained by epitaxial growth of Cu(111) (i.e. Cu(111) film) on annealed c-plane sapphire. Inset: Statistical results of the stacking order (AB stacking or non-AB stacking) (centre) and STEM image of the AB-BLG with atomic resolution (top right). **b** Raman spectra of the AB-BLG transferred onto a SiO₂/Si substrate acquired via a line scanning with the step of 50 μm by normalizing the G band intensity. **c** Raman 2D band of the AB-BLG, which is fitted by four Lorentzian peaks. **d** Two-dimensional plot of the ρ_{xx} as functions of both V_{tg} and V_{bg} of a dual-gate AB-BLG device. **e** Photocurrent distribution in the two graphene/electrode junctions. Inset: OM image of the graphene FET device. Scalebar: 5 μm. **f** Photocurrent mapping result of the graphene device, in which the MLG is on the left part and the BLG is on the right part.

7) Can the authors check if Supplementary Fig. 29d should be for the conductance?

Reply: Thanks for the kind reminding. We have checked Fig. 29d and changed it to conductance (Fig. R10). Fig. R10 has been put in the updated supporting information as Supplementary Fig. 34.

Figure R10. Electrical measurement of the dual gate AB-BLG Hall bar device. **a** Schematic of the Hall bar device of the AB-BLG. **b** OM image of the AB-BLG dual gate Hall bar device. **c** Resistivity of the AB-BLG as a function of V_{tg} , which is measured at room temperature (300 K) with zero V_{bg} . **d** Resistivity of the AB-BLG as a function of the carrier concentration.

8) In general, the quality of the images of the supporting information is low, rendering them useless (see the previous comment about Figure 8 of the supporting information)

Reply: Thanks for the comments to strengthen the presentation of this work. To further improve the image quality and make them more useful, we have conducted the following revisions. Firstly, we have separated Supplementary Fig. 3 into 3 new figures (Fig. R11-13) by taking out the inset OM images so that the contrast differences caused by graphene layer number can be clearly observed. The three figures are used as Supplementary Figs. 3-5 in the updated supporting information. Secondly, for Supplementary Figures containing images (Supplementary Figs. 2, 6, 8, 9, 11, 13, 14, 16, 18, 19, 22, 23, 25, 26, 28 and 29 in the previous version), we have increased resolution from 300 ppi to 1,200 ppi when outputting them from the Adobe Illustrator software. These figures were put in the new supporting information as Supplementary Figs. 2, 9, 13, 15, 17, 18, 20, 22, 23, 27, 28, 30, 31, 33, and 34. Based on these, we believe that these images could provide enough information to support our main text. We will keep this in mind to avoid such issues in our following works.

Figure R11. Impact of the gas atmosphere on the growth behavior of the BLG. **a,b** Impact of the H₂ (a) and CH₄ (b) flow on the BLG coverage. **c,d** Impact of the CO₂ flux on the nucleation density (c) and domain size (d) of the BLG.

Figure R12. Impact of the H₂ flux on the synthesis of the BLG. **a-c** OM images of the graphene films transferred onto SiO₂/Si substrates, which were prepared using 100 sccm (a), 300 sccm (b), and 1000 sccm (c) H₂.

Figure R13. Impact of the CH₄ flux on the synthesis of the BLG. **a-c** OM images of the graphene films

transferred onto SiO₂/Si substrates, which were prepared using 1 sccm (**a**), 5 sccm (**b**), and 20 sccm (**c**) CH₄.

REVIEWERS' COMMENTS

Reviewer #3 (Remarks to the Author):

I would like to express my appreciation for the authors' effort in addressing my previous concerns and for including the raw data used to produce the figures in the manuscript. Based on the provided information, I now recommend the publication of the manuscript with a minor modification.

In view of the raw data included, I would like to point out that the lower limit of the color scale of Figure 4d (currently 1.8 kOhm) seems to be too high. This may lead to a misrepresentation of the transfer curves, making them appear sharper than they actually are. Therefore, I suggest adjusting the lower limit to a value that more accurately reflects the data being presented.

Additionally, in their reply the authors mentioned that the data in Figure R5 corresponds to p-doped graphene. However, upon review, it appears that the graphene is actually n-doped, unlike the data presented in Figure 4d. Furthermore, the mobility of that device appears to be quite low. I hope the authors can address these concerns before finalizing the manuscript.

REVIEWERS' COMMENTS

Reviewer #3 (Remarks to the Author):

I would like to express my appreciation for the authors' effort in addressing my previous concerns and for including the raw data used to produce the figures in the manuscript. Based on the provided information, I now recommend the publication of the manuscript with a minor modification.

In view of the raw data included, I would like to point out that the lower limit of the color scale of Figure 4d (currently 1.8 kOhm) seems to be too high. This may lead to a misrepresentation of the transfer curves, making them appear sharper than they actually are. Therefore, I suggest adjusting the lower limit to a value that more accurately reflects the data being presented.

Additionally, in their reply the authors mentioned that the data in Figure R5 corresponds to p-doped graphene. However, upon review, it appears that the graphene is actually n-doped, unlike the data presented in Figure 4d. Furthermore, the mobility of that device appears to be quite low. I hope the authors can address these concerns before finalizing the manuscript.

Reply: Great thanks to Reviewer #3 for the efforts in reviewing our manuscript and the explicit recommendations for the publication.

Following the reviewer's suggestion, the scale bar in Fig. 4d has been updated by decreasing the lower limit from 1.8 kOhm to 1.4 kOhm to reflect the data being presented more accurately, as shown in Fig. R1. This figure has also been updated in the main text.

Figure R1. Synthesis of AB-stacking bilayer graphene (AB-BLG) on Cu(111) substrates. **a** Distribution of twist angles based on selected area electron diffraction patterns of the BLG grown on the ultraflat single-crystal Cu(111) substrate that was obtained by epitaxial growth of Cu(111) (i.e. Cu(111) film) on annealed c-plane sapphire. Inset: Statistical results of AB-BLG (red) and non-AB stacking BLG (tBLG) (blue) (centre) and scanning transmission electron microscope image of the AB-BLG with atomic resolution (top right). **b** Raman spectra of the AB-BLG transferred onto a SiO₂/Si substrate acquired via a line scanning with the step of 50 μm by normalizing the G band intensity. **c** Raman 2D band of the AB-BLG, which is fitted by four Lorentzian peaks. **d** Two-dimensional plot of the total resistance (ρ_{xx}) as functions of both top gate voltage (V_{tg}) and back gate

voltage (V_{bg}) of a dual-gate AB-BLG device. **e** Photocurrent distribution in the two graphene/electrode junctions. Inset: Optical microscope image of the graphene field effect transistor (FET) device and the boundary of monolayer graphene (MLG) and BLG regions is marked by the white dashed line. Scalebar: 5 μm . The green and purple areas correspond to the MLG and BLG regions respectively and the blue and red curves correspond to the two black dashed lines in the inset, as denoted by the black arrows. **f** Photocurrent mapping result of the graphene device, in which the MLG is on the left part and the BLG is on the right part, where the two black dashed lines in Fig. 4f correspond to the regions marked by black dashed lines in the inset of Fig. 4e and the white dashed line denotes the boundary of MLG and BLG regions.

However, we need to politely point out that the transfer curve shown in Fig. R5 in our previous Response Letter (Fig. R2c in the present Response Letter) is an indication of the p -doping state of our bilayer graphene. To further explain this, we have plotted the energy structure and corresponding transfer curves of n -doped, non-doped, and p -doped graphene in Fig. R2a-b. The doping states of graphene without applying gate voltage can be reflected by the position of the charge neutrality point (CNP) in the transfer curves. That is, CNP is > 0 V for p -doped graphene and CNP is < 0 V for n -doped graphene. In our experiment, since the bilayer graphene (BLG) on SiO_2/Si substrate is highly p -doped, we observed an obvious increase of its sheet resistance by applying positive gate voltage. This relationship between the sheet resistance and gate voltage is like the behaviour in the left part of the orange curve in Fig. R2b, and thus is a clear indication of the p -doped state of our BLG. Furthermore, we have also plotted the curve showing conductivity dependence on the gate voltage to show the electrical gating effect on the p -doped BLG.

Figure R2. Doping analysis of the BLG device on SiO_2/Si substrate. **a** Schematic showing the different doping states of graphene. **b** Schematic showing transfer curves of graphene with different doping states. **c** Transfer curve of our BLG. Inset: Schematic of the Fermi level (E_f) positions under different gate voltage. **d** Conductivity of the BLG as a function of the gate voltage. Inset: Schematic of the Fermi level (E_f) positions under different gate voltage.

Generally, the carrier mobility of AB-BLG is lower than that of monolayer graphene, owing to its parabolic

band structure. The room temperature carrier mobility of over $5,000 \text{ cm}^2\text{V}^{-1}\text{s}^{-1}$ in our work is comparable to the reported results. Considering that the wet transfer process of graphene causes polymer residues, non-uniform strains in graphene, and imperfect hBN/graphene interfaces, we believe that the measured carrier mobility can be further increased after optimizing the graphene transfer and device fabrication process.